# Modeling the Economic Impacts of
# AI Openness Regulation

**Tori Qiu**
Carnegie Mellon University
`toriq@andrew.cmu.edu`

**Benjamin Laufer**
Cornell Tech
`bdl56@cornell.edu`

**Jon Kleinberg**
Cornell University
`kleinberg@cornell.edu`

**Hoda Heidari**
Carnegie Mellon University
`hheidari@andrew.cmu.edu`

## Abstract

Regulatory frameworks, such as the EU AI Act, encourage openness of general-purpose AI models by offering legal exemptions for "open-source" models. Despite this legislative attention on openness, the definition of open-source foundation models remains ambiguous. This paper models strategic interactions among the creator of a general-purpose model (the *generalist*) and the entity that fine-tunes the general-purpose model to a specialized domain or task (the *specialist*), in response to regulatory requirements on model openness. We present a stylized model of the regulator's choice of an open-source definition to evaluate which AI openness standards will establish appropriate economic incentives for developers. Our results characterize market equilibria — specifically, upstream model release decisions and downstream fine-tuning efforts — under various openness regulations and present a range of effective regulatory penalties and open-source thresholds. Overall, we find the model's baseline performance determines when increasing the regulatory penalty vs. the open-source threshold will significantly alter the generalist's release strategy. Our model provides a theoretical foundation for AI governance decisions around openness and enables evaluation and refinement of practical open-source policies.

## 1 Introduction

Foundation models with publicly available weights, such as Llama and Deepseek [42, 13], have the potential to distribute economic benefits and broaden access to AI development. However, developers of these general-purpose models have faced criticism for marketing their products as "open-source" while departing from established open-source principles and concealing essential design decisions [29]. Traditionally, open-source software (OSS) describes publicly available source code with few restrictions on use, modification, and redistribution [24]. Foundation models complicate the standard open-source definition, as they exhibit different degrees of openness, ranging from models with hosted API access to open-weight models to fully open models with public weights, code, data, and no use restrictions [38, 20]. Open-source definitions that transfer principles of OSS to foundation models prohibit any use restrictions or monetization of the model [41] and may require disclosure of all components, including training data [24]. Industry embraces a more lenient definition, where open-weights models released under a non-proprietary license qualify as open-source, even when commercial use restrictions exist (e.g., Llama and Mistral Large [3, 33]).

Regulatory frameworks for foundation models have proposed various definitions of "open-source" that reflect this uncertainty. While the EU AI Act exempts open-source models from disclosure

39th Conference on Neural Information Processing Systems (NeurIPS 2025).

and documentation requirements, it disqualifies any monetized open-source AI products from using exemptions for open-source models [41]. Singapore's generative AI governance framework stipulates that open-source models should provide all source code, documentation, and data required to retrain the model from scratch [17]. Other regulations provide less clarity — the US AI Foundation Model Transparency Act mentions "open source foundation models" as possibly being exempt from data disclosure requirements, but does not define open source [1].

This work offers a game-theoretic model of strategic interactions under various AI openness regulations. A general-purpose technology producer (the *generalist*) decides an openness level for a multi-purpose AI technology. One or more *domain specialist(s)* may then adapt or fine-tune the base model depending on the openness level set by the generalist. Finally, a regulatory threshold moderates the costs associated with different openness levels. We solve the game by providing a closed-form solution for both players' strategies in Section 2. Sections 3 and 4 analyze the sensitivity of the generalist's openness decision, bargaining outcomes, and the resulting fine-tuning investments to changes in the threshold and penalty. Finally, to illustrate the practical application of our model, we show that the model's predicted equilibrium corresponds to real-world release strategies of developers and present takeaways for open-source AI regulation in Section 5.

## 1.1 Related Work

Open-source contributions are often framed as altruistic [4, 6] because conventional valuation methods cannot appraise products with a price of zero and unknown quantities due to unrestricted copying and redistribution rights [22]. However, when developers' decisions to open source are viewed as strategic responses to competitors or internal production constraints, these openness decisions transform into a competitive choice to reduce costs or penetrate price-sensitive markets [39]. Dynamic models of competition and adoption in markets with proprietary software and OSS substitutes [5, 10, 39] link open-source decisions to product performance, capturing profit-driven strategy rather than appealing to altruism. For foundation models in particular, recent work examines how openness affects strategic behavior throughout the AI value chain. Building on the theoretical framework of *fine-tuning games*, Xu et al. [45] show that intermediate levels of openness can motivate early-market specialists to strategically curtail their fine-tuning efforts to outcompete subsequent specialists, and this reduced model performance leaves generalists, specialists, and end consumers worse off relative to zero openness. Wu et al. [44] similarly assess how open-source engagement with a closed-source counterpart impacts innovation for pretrained and adapted foundation models [44].

This paper makes two contributions to modeling strategic openness decisions. First, while existing models evaluate openness as a binary between fully closed or fully open [39, 44], we analyze how generalists strategically adjust their openness levels along a spectrum in order to capture the competitive dynamics of offering partially open models that do not clear the open-source threshold. Second, we consider how administratively imposed openness guidelines alter the generalist's release strategy. Xu et al.'s [45] setup assumes there is a single, exogenous openness level imposed on the generalist, with the analysis centering on how this fixed level affects specialist strategies. Unlike approaches which evaluate the effects of openness on technology adoption and improvement (or fine-tuning) decisions at the specialist level, our setup simultaneously models how generalists adapt their openness and compliance decisions in response to regulatory constraints.

## 2 Model

This section describes our model of strategic interactions between actors who contribute to the development of a general-purpose AI model. We begin by introducing the concept of *openness* we aim to measure, then present the model formally.

## 2.1 Levels of Openness

Our concept of openness, denoted $\omega \in [0, 1]$, projects the observed behaviors of developers onto a continuum to reason about how these behaviors affect developer revenue and strategic reactions. We offer a numeric score for the sake of analysis, under the assumption that the set of choices a firm makes can be mapped into a single range that summarizes the cumulative amount of effort invested in opening a model. We rely on prior work to define how different openness practices correspond to

points on the continuum [38, 9] and treat this one-dimensional measure of openness as a modeling abstraction.

| Fully closed | Hosted access | API access to model | API access to fine-tuning | Weights available | Weights, code, data available with use restrictions | Weights, code, data available without use restrictions |
|---|---|---|---|---|---|---|

$$0 \longleftarrow \text{Model Access} \longrightarrow 1$$

Figure 1: Example of an openness continuum defining $\omega \in [0, 1]$ based on model access [38].

Several works suggest approaches for converting multidimensional openness assessments of models into a spectrum, using feedback mechanisms, risk descriptions, levels of system access, and other indicators to quantify openness [9, 29, 38]. Our work builds on these recommendations by characterizing how the evaluation of openness along such a spectrum affects the incentives of general-purpose model developers.

## 2.2 Game Setup

The interactions between AI developers are formalized as a game between two players, which we refer to as the generalist ($G$) and domain specialist ($D$). Before the game begins, we assume that the generalist has invested in bringing the general-purpose model to an initial performance level $\alpha_0 \in \mathbb{R}^+$. Performance represents a coarse measure of the model's accuracy on target tasks, along with generation speed and efficiency. We further assume that a regulator publishes an open-source threshold $\theta \in [0, 1]$, which is a value on some continuum of openness. The regulator's choice of $\theta$ defines the threshold at which $G$ receives legal exemptions and, by extension, dictates $G$'s costs of development for an open vs. closed model. After the regulation is announced, $G$ and $D$ make investment decisions sequentially, as proposed in Laufer et al. [26]. The generalist must decide an openness level $\omega \in [0, 1]$ to offer the model at given the $\theta$ threshold. An openness level of $\omega = 0$ corresponds to $G$ choosing not to release the model to market, whereas an openness level of $\omega = 1$ indicates a fully open model with all components publicly accessible and zero use restrictions. The specialist then brings the model to an improved performance level $\alpha_1$ in the fine-tuning stage, cooperating with the generalist to increase their joint revenue. The full game with $G$'s openness decision has the following stages:

1. $G$ and $D$ bargain over $\delta(\omega, \alpha_1) \in [0, 1]$, a coefficient that controls the division of revenue between the two players.

2. $G$ chooses an openness level $\omega \in [0, 1]$ for the base model with performance $\alpha_0$, where the model is considered open if $\omega \geq \theta$ and closed otherwise. An openness level closer to $0$ represents a more closed model, whereas $1$ represents a fully open model. When $G$ abstains from releasing the model ($\omega = 0$), the specialist receives zero utility.

3. Assuming $G$ does not abstain, $D$ chooses to adopt $G$'s model and invests in improving it to performance $\alpha_1 \geq \alpha_0$.

We assume that players will choose to abstain when their utilities are negative, although in practice, developers may tolerate short-term losses. The final revenue shared by $G$ and $D$, given by a revenue function $r(\alpha_1) : \mathbb{R}^+ \to \mathbb{R}^+$, depends on the model's final performance $\alpha_1$. For simplicity and unless otherwise stated, we assume $r$ is the identity function in our analysis. In standard cooperative bargaining games [25, 37], $\delta(\omega, \alpha_1)$ represents the result of a linear revenue-sharing contract between players. $G$ receives $\delta(\omega, \alpha_1) \cdot r(\alpha_1)$ as its share of the final revenue, while $D$ receives $(1 - \delta(\omega, \alpha_1)) \cdot r(\alpha_1)$. However, this model of surplus division in bilateral bargains [16, 35] is challenged when an openness decision enters $G$'s strategy. The bargaining parameter $\delta(\omega, \alpha_1)$ represents the result of licensing and subscription arrangements when dividing revenue from foundation model development. Higher openness forces $G$ to forfeit a share of the bargained revenue $\delta(\omega, \alpha_1) \cdot r(\alpha_1)$ because $G$ can no longer charge directly for model usage or other support services and must instead replace direct revenue from model performance with indirect benefits, such as a reputational premium for offering an open model. $\delta(\omega, \alpha_1)$ is simplified to $\delta$ for the remainder of the paper, but we treat it as endogenous to players' strategic decisions due to the bargaining stage.

If the model is open, $G$'s payoff is fixed to $\epsilon\alpha_1$, where $\epsilon \in [0, 1]$ is a constant that captures reputational gains of $G$. The $\epsilon$ constant accounts for "commons-based" production, where participants organize efforts and capture value without exclusive rights via property or contracts, relying instead on indirect rewards like reputation and cost sharing [7]. This contrasts with the $\delta$ parameter, which reflects direct market incentives through a revenue-sharing agreement between the generalist and specialist.

## 2.3 Utility & Cost Functions

| Utility ($U$) | Cost ($\phi$) |
|---|---|
| $U_G := (\epsilon\omega + \delta(1 - \omega)) \cdot r(\alpha_1) - \phi_G(\omega\|\theta)$ | $\phi_G := \alpha_0\omega + c_\omega\alpha_1(1 - \omega) + p\mathbb{1}[\omega < \theta]$ |
| $U_D := (1 - \delta(1 - \omega)) \cdot r(\alpha_1) - \phi_D(\alpha_1\|\omega, \alpha_0)$ | $\phi_D := (\alpha_1 - \alpha_0)^2\omega^{-1} + c_\omega\alpha_1\omega$ |

Table 1: Utility and cost functions for the generalist and specialist. The constant parameters $\epsilon$ and $c_\omega$ define costs and rewards exogenous to the game, and the value of $\alpha_0$ is fixed because our game is only concerned with $G$'s decision to open an existing general-purpose model. Openness regulation takes the form of parameters $\theta$ (representing stringency of the open-source definition) and $p$ (representing the penalty for failing to meet the open-source definition).

Table 1 describes the cost and utility functions of both players. Each player's utility is calculated as their revenue share minus the costs of model improvement and hosting. $G$'s revenue structure changes according to the openness level at which the model is offered, with reputational gains ($\epsilon$) influencing revenue more as openness increases and the bargaining outcome ($\delta$) influencing revenue more as openness decreases.

$$\phi_G(\omega|\theta, \alpha_0) = \underbrace{\alpha_0\omega}_{\text{1. production}} + \underbrace{c_\omega\alpha_1(1 - \omega)}_{\text{2. operation}} + \underbrace{p\mathbb{1}[\omega < \theta]}_{\text{3. regulatory}} \tag{1}$$

**Generalist Cost Function.** $G$'s cost function (Equation 1) can be categorized into production costs, operation costs, and regulatory costs. *Production costs* refer to a fixed cost the generalist faces to offer the model with performance $\alpha_0$ an openness level $\omega$. In our case, we use $\alpha_0\omega$ to cover the production costs of documenting and releasing the model, which scale with higher levels of openness and performance. *Operation costs* capture the variable expenses required to run and host the model, including costs of inference and system maintenance. For a more closed model, $G$ hosts specialized servers to process user requests and charges the user via pay-per-use API calls. The generalist's operation costs decrease with $\omega$ because $G$ can offload hosting and inference costs to users and benefit from community-built optimizations at higher levels of openness. For simplicity, we normalize the coefficients for production and regulatory compliance costs to one, using $c_\omega$ to compare between operation costs and revenue. The constant $c_\omega$ quantifies the ratio of average operation costs to revenue per unit of performance for $D$. $G$ pays an additional *regulatory cost*, $p$, to comply with regulatory requirements when the model is closed ($\omega < \theta$) and does not quality for open-source exemptions.

$$\phi_D(\alpha_1|\omega, \alpha_0) = \underbrace{(\alpha_1 - \alpha_0)^2\omega^{-1}}_{\text{1. production}} + \underbrace{c_\omega\alpha_1\omega}_{\text{2. operation}} \tag{2}$$

**Specialist Cost Function.** The total cost for the domain specialist $\phi_D$ (Equation 2) is composed of production costs and operation costs. As $\omega$ approaches zero, $D$'s production costs go to infinity, indicating an infeasible strategy where it becomes prohibitive to fine-tune or improve a model with heavily restricted access. However, cheaper production costs for models with higher levels of openness are partially offset by the operation costs $D$ must pay for self-hosting the model and shouldering inference costs directly. We prove results for a particular set of quadratic cost functions and monotonic revenue functions, although these represent only a specific subset of possible functional forms for the broader class of utility functions. Specifically, we assume $D$ faces quadratic costs when developing the model from performance $\alpha_0$ to $\alpha_1$ since incremental improvements are more expensive at higher performance, and discount development costs for $D$ by a factor of $\omega^{-1}$ since there are fewer charges associated with adopting and improving a more open model (Table 1).

**Specialist & Generalist Cost Sharing.** In the basic version of the fine-tuning game presented by prior work [26], model performance is the only feature in $G$'s strategy space. The cost functions for the generalist and specialist are condensed into single terms, as $\phi_G = {\alpha_0}^2$ and $\phi_D = (\alpha_1 - \alpha_0)^2$. This simplification is effective when investments in model performance represent a *joint* production process with shared revenue, as development and operation costs rise with performance for the generalist and specialist. However, our game setup explicitly models both performance and openness as decision variables. Unlike the original fine-tuning game [26], where costs scale equivalently for all features (the single feature being performance), openness imposes opposite effects on cost for the two players. For any given $\alpha_1$, higher openness decreases production costs for the specialist but increases operation costs. These changes in cost are shifted to the generalist, and higher openness increases production costs for the model but decreases operation costs for the generalist. To capture how openness induces cost shifts between players, our game explicitly decomposes cost functions into separate terms for development and operation. Appendix C has more detailed discussion of the utility functions and cost sharing.

# 3 Analysis of Equilibrium Strategies

This section characterizes the subgame-perfect equilibria strategies as a function of the game parameters $\epsilon, c_\omega, p, \theta$. We provide closed-form expressions for the best response of each player under a fixed revenue-sharing parameter $\delta$. Proofs of all propositions are provided in Appendix B.1.

## 3.1 Subgame Perfect Equilibria

**Proposition 1** (Characterization $D$'s Equilibrium Strategy). *In a game with quadratic costs and a monotonic revenue function $r(\alpha_1)$, if $\omega \leq \frac{1-\delta}{c_\omega - \delta}$, then $D$'s best-response strategy is given by $\alpha_1^* = \alpha_0 + \frac{\omega(1 - \delta(1 - \omega) - c_\omega \omega)}{2}$. If $\omega > \frac{1-\delta}{c_\omega - \delta}$, $D$ will choose to abstain.*

$D$ must choose a performance level $\alpha_1^* \geq \alpha_0$ for their adapted, or fine-tuned, model in the game. The condition for $D$'s participation in Proposition 1, $\omega \leq \frac{1-\delta}{c_\omega - \delta}$, is only satisfied when the costs of operating the model at baseline performance do not exceed $D$'s share of $r(\alpha_1)$. Using the specialist's best response, $G$'s strategy can then be solved in terms of the $\theta$ threshold.

**Proposition 2** (Characterization of $G$'s Equilibrium Strategy). *In a game with quadratic costs and a monotonic revenue function $r(\alpha_1)$, $G$'s preferred openness $\omega^*$ will be one of the values in the following set:*

$$
\omega^* \in \Bigg\{ \frac{1}{3(\epsilon - \delta + c_\omega)(\delta + c_\omega)} \Big( (\epsilon - \delta + c_\omega)(1 - \delta) + (c_\omega - \delta)(\delta + c_\omega) \pm
$$
$$
\Big[ \big( (\epsilon - \delta + c_\omega)(1 - \delta) + (c_\omega - \delta)(\delta + c_\omega) \big)^2 +
$$
$$
6(\epsilon - \delta + c_\omega)(\delta + c_\omega)\Big( (\epsilon - \delta - 1 + c_\omega)\alpha_0 + \frac{(\epsilon - c_\omega)(1 - \delta)}{2} \Big) \Big]^{\frac{1}{2}} \Big), 0, \theta, 1 \Bigg\}
$$

*$G$ will choose the value in the set which maximizes $U_G$. $G$ will choose to abstain if and only if $U_G < 0$ for all candidates in the set.*

## 3.2 Generalist Strategies at Equilibrium

We begin by characterizing $G$'s openness strategies at equilibrium for a two-player game under different bargaining agreements without regulation ($p = 0$ and $\theta$ has no effect). The Nash bargaining solution calculates joint utility as the product of $U_D$ and $U_G$, the vertical monopoly (VM) solution as the sum, and the egalitarian solution as the minimum of $U_D$ and $U_G$.

**Bargaining Effects.** The $\delta^*$ value which maximizes a specific measure of joint utility is identified via grid search, as an analytical solution for the equilibrium bargaining agreement is intractable with multiple game parameters and $\omega^*$ selected from a set. Figure 2 visualizes $G$ and $D$'s equilibrium strategies over initial model performance ($\alpha_0$) and reputational benefits ($\epsilon$) when no regulation is

imposed. $G$ chooses a fully open model when the bargained revenue-sharing agreement is $\delta^* = 0$, but chooses a fully closed model when $\alpha_0$ is high and $\delta^*$ is moderate. Counterintuitively, $G$ chooses intermediate openness when $\delta^*$ is maximized and the bargained revenue-sharing outcome for a closed model strongly favors $G$. This intermediate openness level empowers $D$ to adopt and improve the model at low $\alpha_0$, which generates greater total revenue than would be possible under a fully closed approach. Full plots of the equilibrium outcomes with variations to $\alpha_0$, $\epsilon$, and $c_\omega$ are in Appendix G.

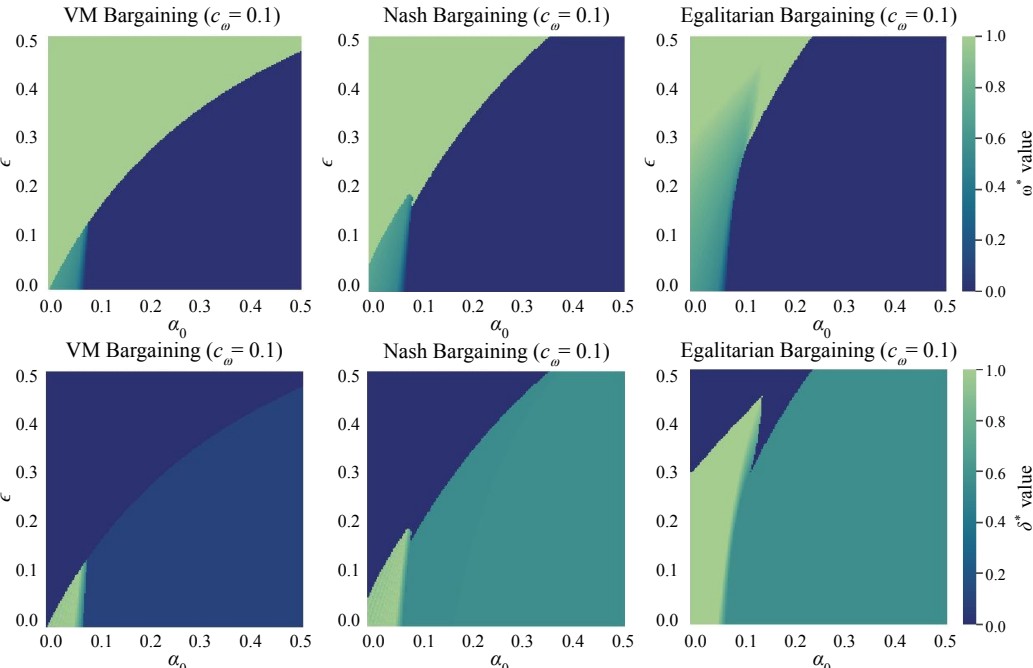

Figure 2: $G$'s equilibrium strategies for $\omega^*$ (top row) and $\delta^*$ (bottom row) at $c_\omega = 0.1$ with no regulation ($p = 0$). Low initial performance ($\alpha_0$) and high reputational benefits ($\epsilon$) lead to partial openness. Since $G$ unilaterally controls the model's release strategy, $G$ can use the openness decision to remove or weaken the bargain, explaining why a fully open model coincides with bargains that allocate no closed revenue to $G$.

In general, $G$ prefers a fully closed strategy as the initial model performance $\alpha_0$ increases, even for higher $\epsilon$ values, because the performance is high enough to generate considerable subscription and licensing revenue via $r(\alpha_1)$ without specialist improvements.

**Openness-Performance Tradeoff.** $G$'s no-regulation equilibrium strategies corroborate empirical patterns of model release, where models with lower performance are relatively more open access and closed models outpace open-weight models by several months in terms of performance (Figure 3). The openness level chosen by $G$ at equilibrium from our main theoretical result quantifies the observed tradeoff between performance and openness in release strategies in terms of $\epsilon$ and $\alpha_0$, as well as the $\delta$ value bargained by players. When $\alpha_0$ and $\epsilon$ are both low, $G$ chooses intermediate openness levels to reduce the specialist's costs because the model is not performant enough to drive adoption at full cost under a closed approach. This outcome corresponds to proprietary decisions to release underperforming models with partial open access, such as Google releasing lower-capability open variants alongside high-performing closed models (Figure 3).

## 4   Analysis of Regulation Effects

To identify effective open-source interventions, we characterize the range of regulatory effects across initial model performances and identifies specific penalty and threshold configurations required to achieve compliance. The space of key constants $\epsilon, \delta, c_\omega$ was systematically probed (details in Appendix G) and in what follows, we establish the existence of three qualitatively different regulatory outcomes.

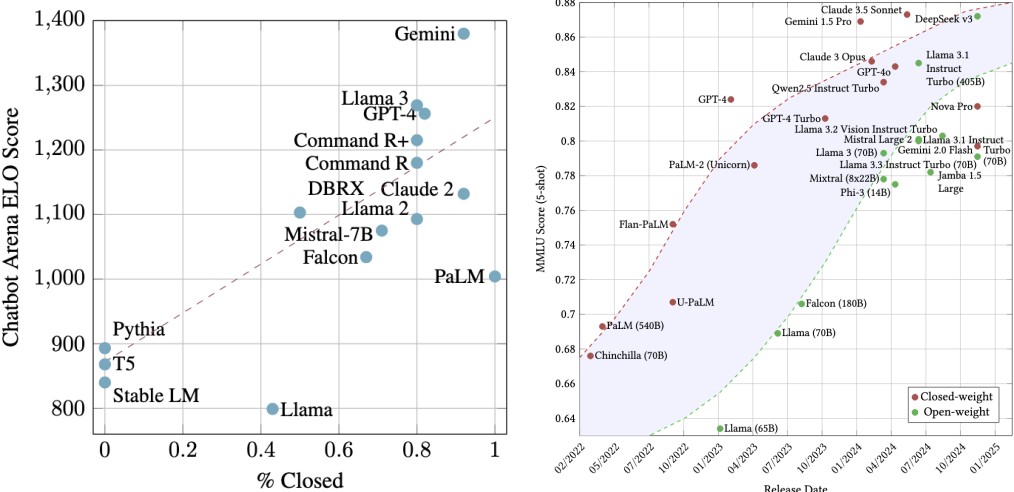

Figure 3: Overview of empirical model release strategies. **(Left)** Higher performance of a model corresponds to a higher percentage of closed components, determined by Eiras et al.'s assessment of model components [14]). **(Right)** For each generation, closed-weight models have higher performance than open-weight ones, despite open-weight models showing comparable performance to closed-weight models from previous time periods. The performance gap between open- vs. closed-weight models (blue region) persists even as absolute performance improves across generations. Appendix F reports the figure data.

## 4.1 Regulation Effects on Openness Decisions

The boundary where $G$'s openness decision transitions from a fully closed model to a partially open one represents the *indifference curve* where $G$'s utility for maintaining a fully closed model ($\omega \approx 0$) is equal to the utility for opening the model. For any $(p, \theta)$ combination below this curve, $G$ will choose an openness level that meets the threshold. We provide a closed-form expression for this curve that identifies which combinations of $(p, \theta)$ will drive changes to $G$'s openness strategy. Proofs are in Appendix E.

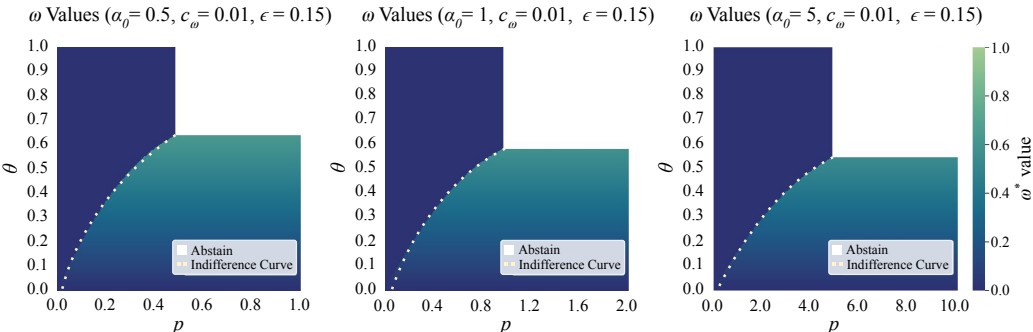

Figure 4: Indifference curves for the generalist over $(p, \theta)$ choices for game parameters $c_\omega = 0.01, \epsilon = 0.15$ and $\alpha_0 \in \{0.5, 1, 5\}$. In the region of non-compliance above the indifference curve, $G$ keeps the model at an openness level $\omega^* \to 0$. In the area of compliance below the indifference curve, $G$ chooses $\omega^* = \theta$.

**Proposition 3.** *The indifference curve separating fully closed from partially open strategies in the space of regulation profiles $(p, \theta)$ is defined by $p = \theta(\delta + \frac{\alpha_0}{\alpha_1} - \epsilon - c_\omega)\alpha_1$. For any $p < \theta(\delta + \frac{\alpha_0}{\alpha_1} - \epsilon - c_\omega)\alpha_1$, $G$ fully closes the model.*

**Regulation can lead to deadweight loss.** The area above the indifference curve illustrates an outcome where openness regulation lowers both players' utilities without yielding any openness

improvements (Figure 4). There are no corresponding improvements to $\alpha_1$ in this region because $D$ cannot contribute via fine-tuning efforts. Above the indifference curve, any increases to the penalty directly deduct from $G$'s utility because the model stays fully closed. $D$'s revenue share is also reduced because the bargain shifts in $G$'s favor as the penalty is increased. Extreme combinations of $(p, \theta)$ do not just produce deadweight loss but cause $G$ to withdraw from the market entirely by abstaining, which represents the worst-case outcome because no positive utility can be generated.

## 4.2 Regulation Effects With Compliance

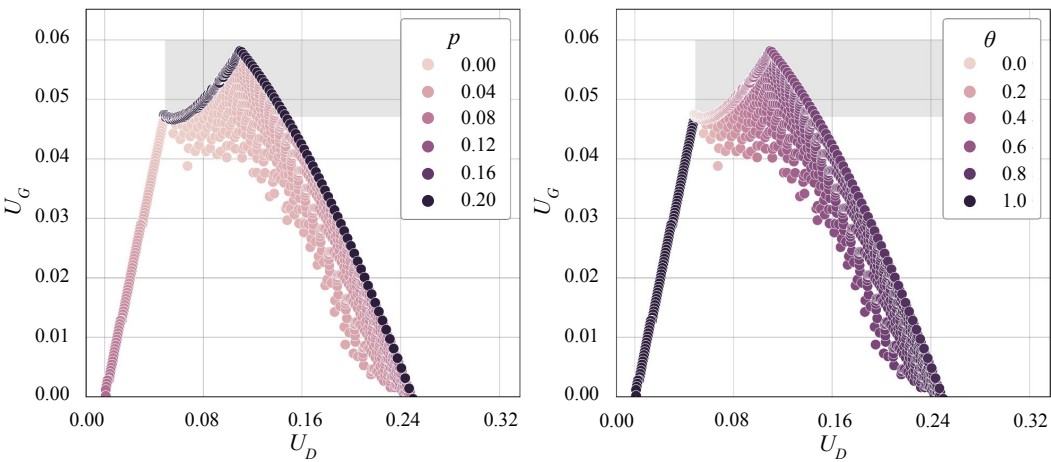

Figure 5: Player utilities under various $(p, \theta)$ regulations when $\alpha_0 = 0.1, c_\omega = 0.05$, and $\epsilon = 0.1$ with Nash bargaining. $(U_G, U_D)$-improving regulations are enclosed in the gray region, with the lower left corner of the gray region corresponding to the no-regulation equilibrium where $p = 0$ and $\theta$ has no effect. For overly stringent $\theta$ thresholds, both players' utilities decrease.

**Definition 1** (Pareto-Optimal Policies). *Given a vector of $d$ utility functions $\mathbf{u} \in \mathbb{R}^d$, the Pareto-optimal policy is the regulatory profile $(\theta, p)$ that maximizes the following optimization problem: $\arg\max_{p,\theta} \ w^\top \mathbf{u} \ s.t. \ \|w\|_1 = 1, w_i > 0, \forall i \in [d]$. The Pareto-optimal policies optimize some weighting over the objectives, where the weight vector $w$ sits on the standard $d$-simplex.*

**Regulation can lead to Pareto improvement over** $(\omega, \alpha_1, U_G, U_D)$**.** When initial performance ($\alpha_0$) is low, there is a region of regulation where $G$ complies with the open-source threshold but would fully close otherwise. This intermediate openness level is mutually beneficial because $D$ makes significant model improvements that increase the total amount of surplus allocated by the bargain. Figure 6 presents one set of parameters where a region of regulation achieves Pareto improvement over $(\omega, \alpha_1, U_G, U_D)$, guiding players toward a mutually beneficial equilibrium that is unachievable without imposing a penalty.

The mutually beneficial openness level is not chosen in the absence of regulation because bargaining alone does not provide a credible commitment mechanism. There is no assurance that $G$ will keep the model open rather than fully closing it to maximize its own $\delta$-share if $D$ concedes a greater revenue share. Cheap shot strategies that are individually beneficial to $G$ are constrained by regulation; the penalty ensures $G$ will at least open the model at the threshold under certain conditions, and $D$ can safely offer a more favorable bargain to secure $G$'s cooperation. In Figure 7, the no-regulation bargaining outcome is roughly balanced ($\delta^* \approx 0.5$), but introducing a penalty $p > 0$ changes the bargaining outcome so that $\delta^* \approx 1$ as the threshold increases below the indifference curve.

**Regulation can encourage specialist innovation.** Regulation that falls below the indifference curve redistributes utility from $G$ to $D$ when $\alpha_0$ is high enough for $G$ to sacrifice licensing and subscription revenue by choosing higher levels of openness. Figure 7 illustrates how increasing the open-source threshold below the indifference curve transfers utility from $G$ to $D$ without deterring $G$'s participation. Higher openness levels in this region raise $\alpha_1$ by reducing production costs for $D$, but $G$ loses closed revenue. For instance, Meta loses licensing revenue by developing and releasing

open-weight models, but cheap access to advanced models allows specialists to develop derivatives like Vicuna and Alpaca [11, 40] which can surpass the original Llama models in performance.

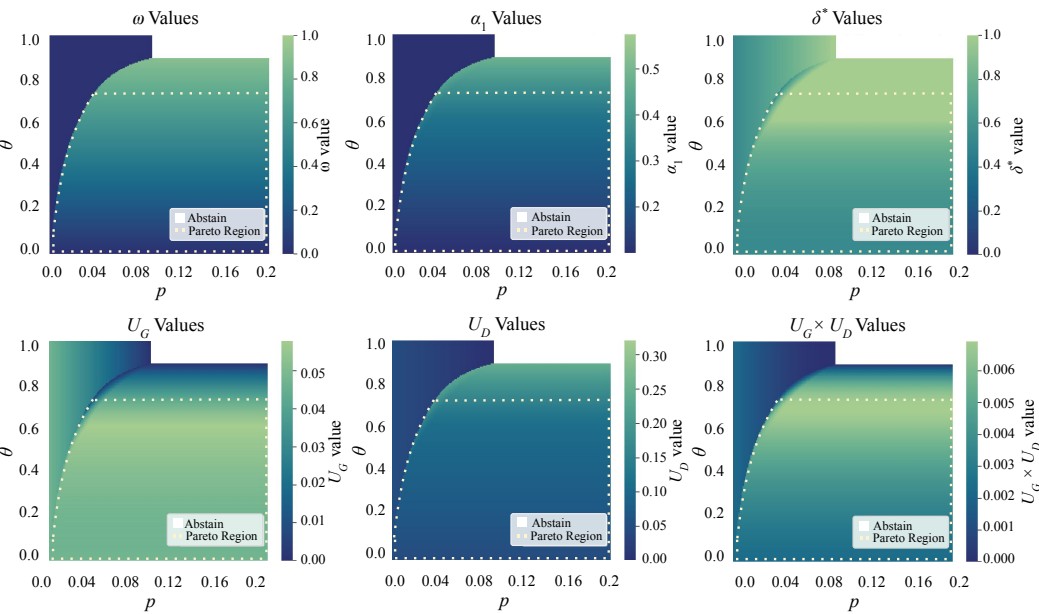

Figure 6: Equilibrium outcomes for $\alpha_0 = 0.1, c_\omega = 0.01, \epsilon = 0.1$ with Nash bargaining show Pareto improvement over utilities $(\omega, \alpha_1, U_G, U_D)$ for any $(p, \theta)$ regulation in the dotted region. At the no-regulation equilibrium ($p = 0$), players' decisions are $\omega \approx 0$ and $\alpha_1 = 0.102$.

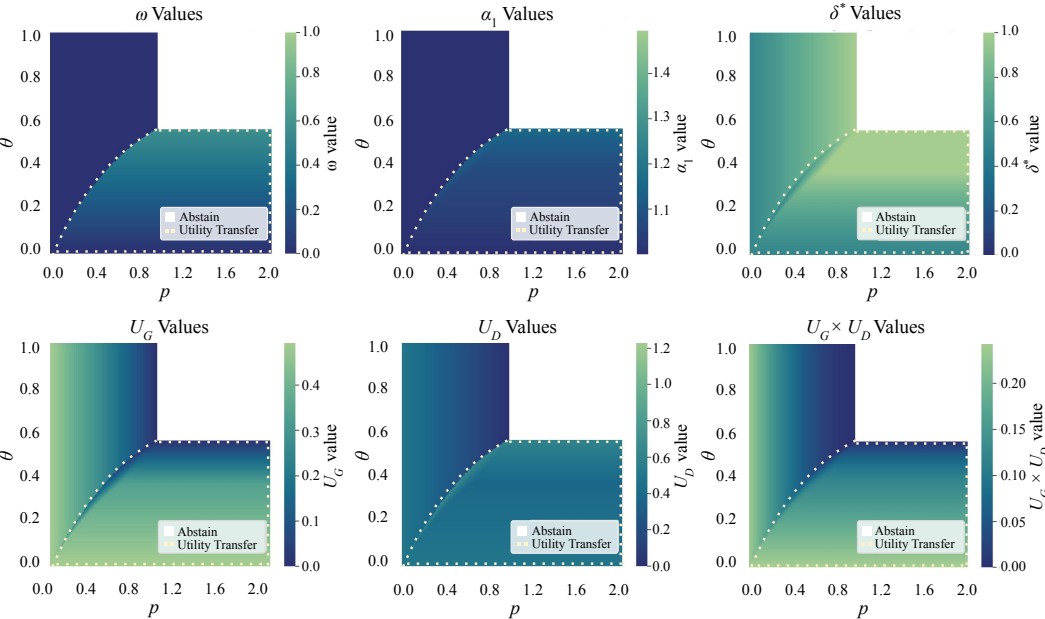

Figure 7: Equilibrium outcomes for $\alpha_0 = 1, c_\omega = 0.01, \epsilon = 0.1$ with Nash bargaining create a region where raising the open-source threshold improves $\omega$, $\alpha_1$, and $U_D$ but lowers $U_G$. This utility transfer encourages specialist innovation, measured by $\alpha_1$. At the no-regulation equilibrium ($p = 0$), players' decisions are $\omega \approx 0.01$ and $\alpha_1 = 1$.

# 5 Discussion

This work introduces an openness decision into the generalist's strategy space, modeling how this parameter affects strategic equilibria in foundation model development. While existing models of openness regulation assume compliance is mandatory and all model developers make homogeneous openness decisions [45], our setup allows generalists to select a continuous openness level based on regulation intensity and model performance. The model we present therefore explains the strategic positioning of AI models along an openness-performance frontier and anticipates how adjustments to open-source thresholds will impact fine-tuning and bargaining dynamics.

## 5.1 Implications for Open-Source AI Regulation

**Regulatory approaches to openness should shift with average model performance to empower innovation.** Low initial performance creates opportunities for Pareto improvements through openness regulation, while high performance at most permits the strategic redistribution of utility between generalists and specialists. As model capabilities evolve, regulatory strategies may need to shift to redistribution approaches that target the specialist's willingness to contribute to model performance. This targeted approach can address competitive imbalances between generalists and specialists, even in environments where a few generalists dominate the offerings of high-performance foundation models. Given that models may excel in certain domains while underperforming in others, regulators must also determine whether to use composite performance metrics, domain-specific benchmarks, or capability-weighted averages to define model performance, with each approach reflecting different model development priorities.

**Effective open-source thresholds require calibrated penalties.** Regulators should consider enforcement costs in regions where higher thresholds cannot influence the generalist's openness decision. For regions above the indifference curve, higher thresholds cause wasted enforcement costs on evaluation and monitoring of additional closed model components, without any corresponding improvements to model openness. Resources allocated to open-source regulation should be concentrated near the indifference curve, and enforcement approaches should scale the regulatory penalty based on specific model characteristics like baseline performance or operation costs to ensure that all generalists face real compliance incentives. This prevents scenarios where developers of high-performing models absorb regulatory penalties as a business expense rather than modifying their transparency practices.

## 5.2 Limitations and Future Work

Our model abstracts from several behavioral and competitive dynamics that can affect the theoretical results presented in this paper. For example, regulatory thresholds can create psychological anchoring that influences behavior independent of the penalty, as firms target the threshold even if the penalty is minimal. Firms may also feel competitive pressure to meet the threshold if they believe that competitors will meet it, creating norms that transcend direct cost calculation of the penalty. Modeling multi-firm competition with a closed-source incumbent can also reveal when established firms pivot to open-source strategies and how timing of market entry affects whether entrants adopt open-source approaches.

The bargaining process may deviate similarly from axiomatic solutions. Our analysis mainly examines Nash bargaining, which uniquely satisfies independence of utility origins and units, Pareto efficiency, symmetry, and independence of irrelevant alternatives [31]. The two alternative bargaining solutions we consider (VM and Egalitarian) are also Paretian, symmetric, and independent of utility origins. However, generalists and specialists may arrive at non-standard bargaining agreements in practice which violate these properties, especially when utilities are expressed in currency and players have bounded rationality [15].

While the main analysis assumes that higher model openness and performance are generally beneficial, a regulator also considers safety risks associated with higher performance or market-wide loss scenarios, such as intermediate levels of openness where consumers may be worse off [45]. This complexity suggests that defining openness as a multidimensional property may be valuable to capture factors beyond simple model access, such as documentation, licensing, and explainability of model outputs [28].

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

# A  Additional Related Work

**Models of AI Regulation.** Numeric thresholds dictate in practice which "covered" products are subject to regulatory standards. The number of floating point operations (FLOPs), the compute cost based on the average market price of the cloud compute, net annual sales, and the number of active monthly users appear as representative thresholds in AI-related legal measures [2]. Although the accuracies of these thresholds are disputed [21], a binary qualification system for rewards may yield more aggregate effort than a continuous reward structure or rewards based on relative performance, as incentives to improve performance intensify with proximity to the threshold [19]. Existing works on the effect of AI requirements on market dynamics model how firms strategically respond to targeted safety regulations [27] and how reputational pressures to satisfy safety constraints facilitate entry of new developers [32].

**Openness & Market Strategy.** Technology providers can gain a competitive advantage by differentiating through price and performance using one of two strategies: versioning and portfolio broadening [18]. Product versioning refers to the release of multiple iterations of a product within the same market segment, typically by developing closed models that compete primarily on performance quality. In contrast, portfolio broadening involves expanding product scope to cover several market segments, such as introducing cheaper, open variants of models. Developers of open models can acquire greater market share by monetizing paid premium models and services at a later stage (e.g., tiered pricing and commercial licensing) or by inflating demand for complementary goods, using approaches analogous to product seeding or loss leading [34].

**Foundation Model Economics.** Paid, pretrained base models with distributed fine-tuning efforts create a tiered market structure that constrains downstream improvements based on access terms set by the generalist. For example, fully contractible token allocations give the specialist full freedom over how to allocate a budget to fine-tuning, whereas per-task token pricing limits the specialist's fine-tuning options [8]. Theoretical models of foundation model development assess how this production structure affects regulatory efforts [27, 23] and market competition [36, 30], while complementary works focused on provider-side strategy derive pricing recommendations for API endpoints and training data [8, 43, 46] that align the incentives of model developers with contributors.

# B  Derivation of Subgame Perfect Equilibria

## B.1  Solution for Fixed Initial Performance

The game is solved by backward induction using the functions from Table 1.

*Proof of Proposition 1.* First, $U_D$ is strictly concave.

$$\frac{\partial^2 U_D}{\partial \alpha_1{}^2} = \frac{\partial^2}{\partial \alpha_1{}^2}\Big((1-\delta(1-\omega))\underbrace{r(\alpha_1)}_{r(\alpha_1)=\alpha_1} -(\alpha_1-\alpha_0)^2\omega^{-1} - c_\omega\alpha_1\omega\Big)$$

$$= \frac{\partial}{\partial \alpha_1}\Big(1-\delta(1-\omega)-2(\alpha_1-\alpha_0)\omega^{-1}-c_\omega\omega\Big)$$

$$= -2\omega^{-1}.$$

$\frac{\partial^2 U_D}{\partial \alpha_1{}^2} < 0$ because $\omega \in (0,1]$ by definition. Since any local maximum is a global maximum, the critical point of $U_D$ gives $D$'s best response.

$$\alpha_1^* = \operatorname*{argmax}_{\alpha_1} U_D(\alpha_0)$$

$$\Rightarrow \left. \frac{\partial U_D}{\partial \alpha_1} \right|_{\alpha_1 = \alpha_1^*} = 0$$

$$\Rightarrow \left. \frac{\partial}{\partial \alpha_1} \left( (1 - \delta(1-\omega)) \underbrace{r(\alpha_1)}_{r(\alpha_1) = \alpha_1} - \phi_D(\alpha_1 | \omega, \alpha_0) \right) \right|_{\alpha_1 = \alpha_1^*} = 0$$

$$\Rightarrow \left. \frac{\partial}{\partial \alpha_1} \left( (1 - \delta(1-\omega))\alpha_1 - (\alpha_1 - \alpha_0)^2 \omega^{-1} - c_\omega \alpha_1 \omega \right) \right|_{\alpha_1 = \alpha_1^*} = 0$$

$$\Rightarrow 1 - \delta(1-\omega) - 2(\alpha_1^* - \alpha_0)\omega^{-1} - c_\omega \omega = 0$$

$$\Rightarrow \frac{\omega(1 - \delta(1-\omega) - c_\omega \omega)}{2} + \alpha_0.$$

When the condition $\omega \leq \frac{1-\delta}{c_\omega - \delta}$ is satisfied, the critical point is feasible and optimal. In cases where the condition is not satisfied, $U_D$ is negative in all cases. Using the fact that $U_D$ is strictly concave, if the critical point is negative, the numerical optimum of $U_D$ must be at the constraint $\alpha_1 = \alpha_0$.

$$U_D(\alpha_1 = \alpha_0) = (1 - \delta(1-\omega))\alpha_0 - 0 - c_\omega \alpha_0 \omega$$
$$= \alpha_0 \left( 1 - \delta + \delta\omega - c_\omega \omega \right)$$

This utility is non-negative if and only if the condition is met: $(1 - \delta + \delta\omega - c_\omega \omega) \geq 0 \iff \omega \leq \frac{1-\delta}{c_\omega - \delta}$. $D$ will prefer to abstain as long as the condition is not met, and the complete strategy is given by:

$$\text{D's strategy:} \begin{cases} \alpha_1^* = \alpha_0 + \frac{\omega(1 - \delta(1-\omega) - c_\omega \omega)}{2} & \text{if } \omega \leq \frac{1-\delta}{c_\omega - \delta}, \\ \text{abstain} & \text{else.} \end{cases}$$

When $D$ does not abstain, the performance improvement is always non-negative (i.e., no player will strategically invest in degrading performance) because $\frac{\omega(1 - \delta(1-\omega) - c_\omega \omega)}{2} \geq 0 \iff \omega \leq \frac{1-\delta}{c_\omega - \delta}$. $\quad \square$

*Proof of Proposition 2.* Solve for $G$'s best response using $\alpha_1^*$. $G$'s optimization problem becomes:

$$\omega^* = \operatorname*{argmax}_{\omega} U_G(\alpha_0, \omega)$$

$$\Rightarrow \left. \frac{\partial U_G}{\partial \omega} \right|_{\omega = \omega^*} = 0$$

$$\Rightarrow \left. \frac{\partial}{\partial \omega} \left( (\epsilon\omega + \delta(1-\omega))r(\alpha_1) - (\alpha_0 \omega + c_\omega \alpha_1(1-\omega) + p\mathbb{1}[\omega < \theta]) \right) \right|_{\omega = \omega^*} = 0$$

$$\Rightarrow \left. \frac{\partial}{\partial \omega} \left( (\epsilon\omega + \delta(1-\omega))\alpha_1 - \alpha_0 \omega - c_\omega \alpha_1(1-\omega) \right) \right|_{\omega = \omega^*} = 0$$

$$\Rightarrow \frac{\partial}{\partial \omega} \left( (\epsilon\omega + \delta(1-\omega)) \left( \frac{\omega(1 - \delta(1-\omega) - c_\omega \omega)}{2} \right) + \alpha_0 \right) - \alpha_0 \omega - $$

$$\left. (c_\omega - c_\omega \omega) \left( \frac{\omega(1 - \delta(1-\omega) - c_\omega \omega)}{2} \right) + \alpha_0 \right) \right|_{\omega = \omega^*} = 0$$

$$\Rightarrow (\epsilon - \delta)(\frac{\omega^* - \delta\omega^* - (\delta + c_\omega)\omega^{*2}}{2} + \alpha_0) + ((\epsilon - \delta)\omega^* + \delta)(\frac{1 - \delta}{2} - (\delta + c_\omega)\omega^*) - \alpha_0 +$$

$$c_\omega(\frac{\omega^* - \delta\omega^* - \delta\omega^{*2} - c_\omega\omega^{*2}}{2} + \alpha_0) - (c_\omega - c_\omega\omega^*)(\frac{1 - \delta}{2} - (\delta + c_\omega)\omega^*) = 0$$

$$\Rightarrow (\epsilon - \delta)(\frac{\omega^* - \delta\omega^* - (\delta + c_\omega)\boldsymbol{\omega^{*2}}}{2} + \alpha_0) + \frac{(\epsilon - \delta)(1 - \delta)\omega^*}{2} - (\epsilon - \delta)(\delta + c_\omega)\boldsymbol{\omega^{*2}} + \frac{\delta(1 - \delta)}{2} -$$

$$\delta(\delta + c_\omega)\omega^* - \alpha_0 + c_\omega(\frac{\omega^* - \delta\omega^* - (\delta + c_\omega)\boldsymbol{\omega^{*2}}}{2} + \alpha_0) - \frac{(c_\omega - c_\omega\omega^*)(1 - \delta)}{2} +$$

$$c_\omega(\delta + c_\omega)\omega^* - c_\omega(\delta + c_\omega)\boldsymbol{\omega^{*2}} = 0$$

$$\Rightarrow \Big(\frac{-(\epsilon - \delta)(\delta + c_\omega)}{2} - (\epsilon - \delta)(\delta + c_\omega) - \frac{c_\omega(\delta + c_\omega)}{2} - c_\omega(\delta + c_\omega)\Big)\omega^{*2} + (\epsilon - \delta)(\frac{(1 - \delta)\boldsymbol{\omega^*}}{2} +$$

$$\alpha_0) + \frac{(\epsilon - \delta)(1 - \delta)\boldsymbol{\omega^*}}{2} + \frac{\delta(1 - \delta)}{2} - \delta(\delta + c_\omega)\boldsymbol{\omega^*} - \alpha_0 + c_\omega(\frac{(1 - \delta)\boldsymbol{\omega^*}}{2} + \alpha_0) - \frac{c_\omega(1 - \delta)}{2} +$$

$$\frac{c_\omega\boldsymbol{\omega^*}(1 - \delta)}{2} + c_\omega(\delta + c_\omega)\boldsymbol{\omega^*} = 0$$

$$\Rightarrow \Big(-\frac{3}{2}(\epsilon - \delta + c_\omega)(\delta + c_\omega)\Big)\omega^{*2} + \Big((\epsilon - \delta + c_\omega)(1 - \delta) + (c_\omega - \delta)(\delta + c_\omega)\Big)\omega^* +$$

$$(\epsilon - \delta - 1 + c_\omega)\alpha_0 + \frac{(\delta - c_\omega)(1 - \delta)}{2}.$$

Solving for the roots with coefficients $A = -\frac{3}{2}(\epsilon - \delta + c_\omega)(\delta + c_\omega)$, $B = (\epsilon - \delta + c_\omega)(1 - \delta) + (c_\omega - \delta)(\delta + c_\omega)$, and $C = (\epsilon - \delta - 1 + c_\omega)\alpha_0 + \frac{(\delta - c_\omega)(1 - \delta)}{2}$ gives candidate values of $\omega^*$. Since $U_G$ is differentiable between the intervals $[0, \theta]$ and $[\theta, 1]$ when $p > 0$, each of these critical points must be evaluated individually. For some sufficiently small positive value $\mu > 0$, $G$ will prefer $\omega = \theta$ to $\omega = \theta - \mu$. It suffices to show $\lim_{\mu \searrow 0} U_G(\omega = \theta - \mu) < U_G(\omega = \theta)$. Since $U_G = (\epsilon\omega + \delta(1 - \omega))\alpha_1 - \alpha_0^2 - \alpha_0\omega - c_\omega\alpha_1(1 - \omega) - p\mathbb{1}[\omega < \theta]$:

$$\lim_{\mu \searrow 0} U_G(\omega = \theta - \mu) = [const] - p$$

$$\lim_{\mu \searrow 0} U_G(\omega = \theta) = [const]$$

Thus, we do not have to check the upper boundary of the closed interval in the optimization over $\omega$. $\qquad \square$

## C   Robustness Checks

### C.1   Fixed Initial Performance with Specialist-Side Penalty

The main results in Section 3 rely on cost functions that assign the entire regulatory burden to the generalist. However, some regulatory frameworks may also impose disclosure and safety testing requirements on the specialist for fine-tuning a closed model. For instance, EU AI Act Article 25 subjects entities making a "substantial modification" to a high-risk AI system to quality management and documentation requirements. To determine whether the policy recommendations remain valid under this alternative liability structure, we test the scenario where both generalists and specialists face the same regulatory cost $p$. The specialist's cost function with a penalty is $\phi_D := (\alpha_1 - \alpha_0)^2\omega^{-1} + c_\omega\alpha_1\omega + p\mathbb{1}[\omega < \theta]$.

**Proposition 4.** *In a game with quadratic costs, a monotonic revenue function $r(\alpha_1)$, and a specialist-side penalty $p$, if $\omega \leq \frac{1}{\delta - c_\omega}(\frac{p}{\alpha_0} + \delta - 1)$, then D's best-response strategy is given by $\alpha_1^* = \alpha_0 + \frac{\omega(1 - \delta(1 - \omega) - c_\omega\omega)}{2}$. If $\omega > \frac{1}{\delta - c_\omega}(\frac{p}{\alpha_0} + \delta - 1)$, D will choose to abstain.*

*Proof.* D's utility function differs from the one in B.1 only by the constant penalty term $-p\mathbb{1}[\omega < \theta]$. Since this constant disappears when deriving with respect to $\alpha_1$, the strict concavity of $U_D$ and the

derivation of $D$' best response are identical to Proposition 1. Specifically, the first-order condition yields $\alpha_1{}^* = \alpha_0 + \frac{\omega(1-\delta(1-\omega)-c_\omega\omega)}{2}$. The main difference is the feasibility condition, as the utility at the constraint instead becomes:

$$U_D(\alpha_1 = \alpha_0) = (1 - \delta(1 - \omega))\alpha_0 - 0 - c_\omega\alpha_0\omega - p$$
$$= \alpha_0\left(1 - \delta + \delta\omega - c_\omega\omega\right) - p.$$

When the condition $\omega \leq \frac{1}{\delta-c_\omega}(\frac{p}{\alpha_0} + \delta - 1)$ is satisfied, the critical point is feasible and optimal. Since $U_D$ is strictly concave, if the critical point is negative, the numerical optimum of $U_D$ must be at the constraint $\alpha_1 = \alpha_0$. This utility is non-negative if and only if the condition is met: $\alpha_0(1 - \delta + \delta\omega - c_\omega\omega) - p \geq 0 \iff \omega \leq \frac{1}{\delta-c_\omega}(\frac{p}{\alpha_0} + \delta - 1)$. Therefore, $D$'s complete strategy is given by:

$$\text{D's strategy:} \begin{cases} \alpha_1^* = \alpha_0 + \frac{\omega(1-\delta(1-\omega)-c_\omega\omega)}{2} & \text{if } \omega \leq \frac{1}{\delta-c_\omega}(\frac{p}{\alpha_0} + \delta - 1), \\ \text{abstain} & \text{else.} \end{cases}$$

$\square$

$G$'s best response with a specialist-side penalty is the same as in Proposition 2, since introducing the specialist penalty only affects when $D$ abstains, not the solution for $\alpha_1{}^*$. We replicate the equilibrium outcomes using the same game parameters as Figures 6 and 7 with the specialist-side penalty for comparison. Figure 8 (corresponding to Figure 6) and 9 (corresponding to Figure 7) show the following differences:

1. $D$ abstains for a broader range of $(\theta, p)$ combinations above the indifference curve. Specifically, the maximum penalty where $D$ stops participating participating is lower when they adapt a closed model that fails to meet the threshold.

2. The additional penalty term slightly reduces $D$'s utility payoff for choosing $\alpha_1{}^*$, but the reduction is negligible.

The specialist-side penalty results in the same qualitative outcomes outside of abstain regions. Below the indifference curve, lower-performing models generate Pareto improvements, while higher-performing models produce utility transfers under greater openness.

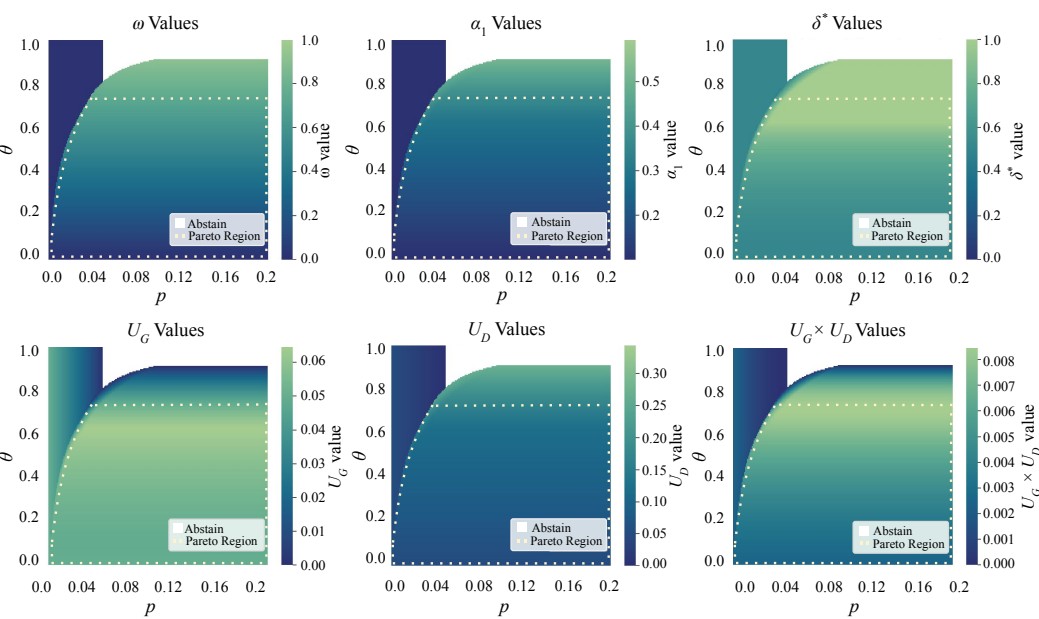

Figure 8: **(Specialist Penalty)** Under a specialist penalty, equilibrium outcomes for $\alpha_0 = 0.1, c_\omega = 0.01, \epsilon = 0.1$ with Nash bargaining show Pareto improvement over utilities $(\omega, \alpha_1, U_G, U_D)$ for any $(p, \theta)$ regulation in the dotted region.

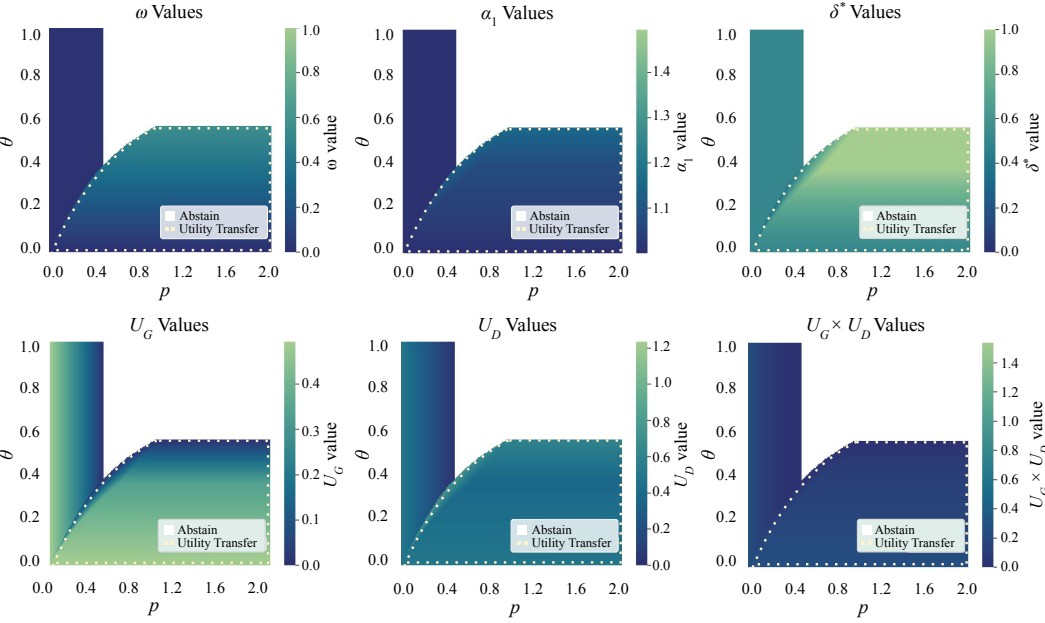

Figure 9: **(Specialist Penalty)** Under a specialist penalty, equilibrium outcomes for $\alpha_0 = 1, c_\omega = 0.01, \epsilon = 0.1$ with Nash bargaining create a region where raising the open-source threshold improves $\omega, \alpha_1,$ and $U_D$ but lowers $U_G$. This utility transfer encourages specialist innovation, measured by $\alpha_1$.

## C.2 Log Scaling for Production Costs

As an additional robustness check, we consider alternative forms of the production costs associated with $G$ releasing the model publicly. Although quadratic revenue remains the dominant term, we explore sublinear scaling for $G$'s production costs. Fixed costs in internal documentation and release

processes suggest that logarithmic scaling may offer a more precise estimate of production costs than linear scaling. With logarithmic production costs ($\alpha_0 \log \omega$), $G$'s optimization problem becomes:

$$\Rightarrow (\epsilon - \delta)(\frac{\omega^* - \delta\omega^* - (\delta + c_\omega)\omega^{*2}}{2} + \alpha_0) + ((\epsilon - \delta)\omega^* + \delta)(\frac{1 - \delta}{2} - (\delta + c_\omega)\omega^*) - \boldsymbol{\alpha_0 \omega^{-1}} +$$

$$c_\omega(\frac{\omega^* - \delta\omega^* - \delta\omega^{*2} - c_\omega\omega^{*2}}{2} + \alpha_0) - (c_\omega - c_\omega\omega^*)(\frac{1 - \delta}{2} - (\delta + c_\omega)\omega^*) = 0$$

$$\Rightarrow (\epsilon - \delta)(\frac{\omega^* - \delta\omega^* - (\delta + c_\omega)\boldsymbol{\omega^{*2}}}{2} + \alpha_0) + \frac{(\epsilon - \delta)(1 - \delta)\omega^*}{2} - (\epsilon - \delta)(\delta + c_\omega)\boldsymbol{\omega^{*2}} + \frac{\delta(1 - \delta)}{2} -$$

$$\delta(\delta + c_\omega)\omega^* - \boldsymbol{\alpha_0 \omega^{-1}} + c_\omega(\frac{\omega^* - \delta\omega^* - (\delta + c_\omega)\boldsymbol{\omega^{*2}}}{2} + \alpha_0) - \frac{(c_\omega - c_\omega\omega^*)(1 - \delta)}{2} +$$

$$c_\omega(\delta + c_\omega)\omega^* - c_\omega(\delta + c_\omega)\boldsymbol{\omega^{*2}} = 0$$

$$\Rightarrow \Big( \frac{-(\epsilon - \delta)(\delta + c_\omega)}{2} - (\epsilon - \delta)(\delta + c_\omega) - \frac{c_\omega(\delta + c_\omega)}{2} - c_\omega(\delta + c_\omega) \Big)\omega^{*2} + (\epsilon - \delta)(\frac{(1 - \delta)\boldsymbol{\omega^*}}{2} +$$

$$\alpha_0) + \frac{(\epsilon - \delta)(1 - \delta)\boldsymbol{\omega^*}}{2} + \frac{\delta(1 - \delta)}{2} - \delta(\delta + c_\omega)\boldsymbol{\omega^*} - \boldsymbol{\alpha_0 \omega^{-1}} + c_\omega(\frac{(1 - \delta)\boldsymbol{\omega^*}}{2} + \alpha_0) - \frac{c_\omega(1 - \delta)}{2} +$$

$$\frac{c_\omega\boldsymbol{\omega^*}(1 - \delta)}{2} + c_\omega(\delta + c_\omega)\boldsymbol{\omega^*} = 0$$

$$\Rightarrow \Big( -\frac{3}{2}(\epsilon - \delta + c_\omega)(\delta + c_\omega) \Big)\omega^{*2} + \Big( (\epsilon - \delta + c_\omega)(1 - \delta) + (c_\omega - \delta)(\delta + c_\omega) \Big)\omega^* +$$

$$(\epsilon - \delta - 1 + c_\omega)\alpha_0 + \frac{(\delta - c_\omega)(1 - \delta)}{2} + \boldsymbol{\alpha_0 \omega^{-1}}$$

$$\Rightarrow \Big( -\frac{3}{2}(\epsilon - \delta + c_\omega)(\delta + c_\omega) \Big)\omega^{*3} + \Big( (\epsilon - \delta + c_\omega)(1 - \delta) + (c_\omega - \delta)(\delta + c_\omega) \Big)\omega^{*2} +$$

$$\Big( (\epsilon - \delta + c_\omega)\alpha_0 + \frac{(\delta - c_\omega)(1 - \delta)}{2} \Big)\omega^* - \boldsymbol{\alpha_0}.$$

Solving the cubic with the following coefficients and checking boundary points gives candidate values of $\omega^*$:

- $A = -\frac{3}{2}(\epsilon - \delta + c_\omega)(\delta + c_\omega)$,
- $B = (\epsilon - \delta + c_\omega)(1 - \delta) + (c_\omega - \delta)(\delta + c_\omega)$,
- $C = (\epsilon - \delta + c_\omega)\alpha_0 + \frac{(\delta - c_\omega)(1 - \delta)}{2})$,
- $D = -\alpha_0$.

With linear scaling for production costs, $G$'s best response is obtained from the set of candidate values given by $A\omega^2 + B\omega + C$ (Appendix B), where $A = -\frac{3}{2}(\epsilon - \delta + c_\omega)(\delta + c_\omega)$, $B = (\epsilon - \delta + c_\omega)(1 - \delta) + (c_\omega - \delta)(\delta + c_\omega)$, and $C = (\epsilon - \delta - 1 + c_\omega)\alpha_0 + \frac{(\delta - c_\omega)(1 - \delta)}{2}$.

Log scaling for production costs produces qualitatively similar outcomes to the main results, as the quadratic or cubic terms from revenue dominate the production cost in either case. We reproduce equilibrium outcomes using the same parameters as Figures 6 and 7 in Figures 10 and 11. For low performance ($\alpha_0 = 0.1$ in Figure 6), $U_G$ increases more due to reduced release costs and the model being significantly more open, but all the core qualitative insights of the analysis are preserved.

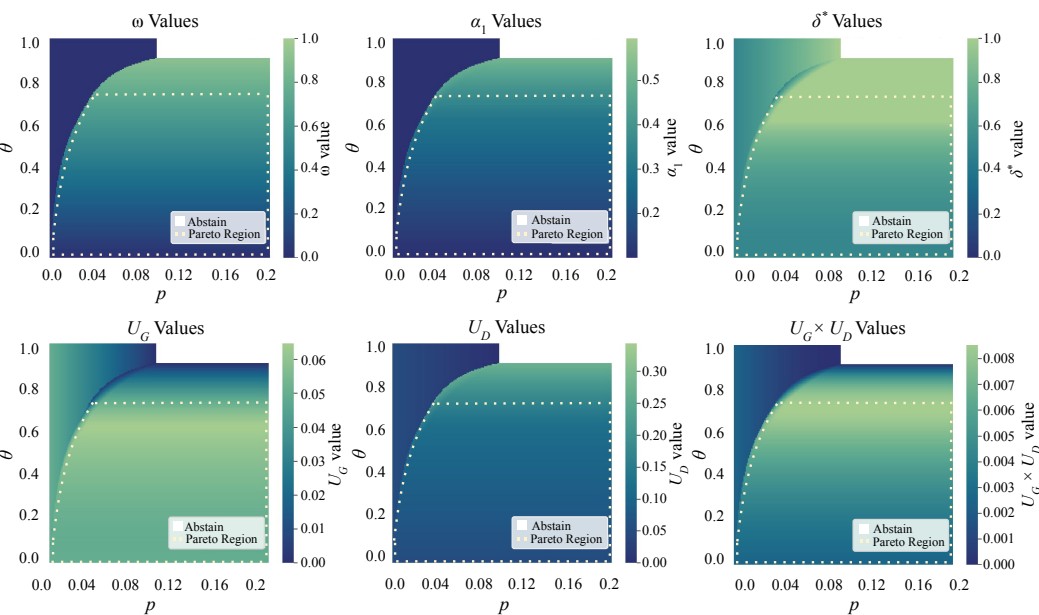

Figure 10: **(Log Scaling)** Under log scaling for production costs, equilibrium outcomes for low performance ($\alpha_0 = 0.1, c_\omega = 0.01, \epsilon = 0.1$) with Nash bargaining still create a region where raising the open-source threshold improves $\omega$, $\alpha_1$, and $U_D$ but lowers $U_G$.

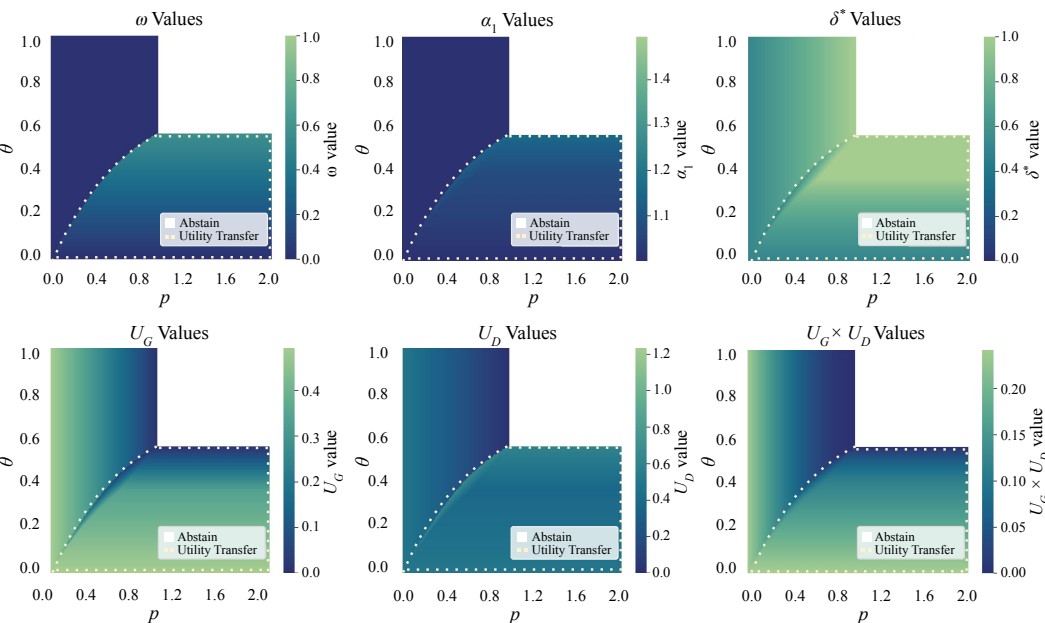

Figure 11: **(Log Scaling)** Under log scaling for production costs, equilibrium outcomes for high performance ($\alpha_0 = 1, c_\omega = 0.01, \epsilon = 0.1$) with Nash bargaining show Pareto improvement over utilities ($\omega, \alpha_1, U_G, U_D$) for any $(p, \theta)$ regulation in the dotted region.

## C.3 Alternate Division of Operation Costs

At either extreme of openness ($\omega = 0$ or $\omega = 1$), either the specialist or generalist will assume the majority of operation costs. For the main results in Section 3, $G$'s operation cost is $c_\omega \alpha_1 (1 - \omega)$ in order to ensure that operation costs sum to a constant between the generalist and specialist:

$\phi_{G_{\text{op.}}} + \phi_{D_{\text{op.}}} = c_\omega \alpha_1 (1 - \omega) + c_\omega \alpha_1 \omega = c_\omega \alpha_1$. Operation costs between $G$ and $D$ sum to a constant value because at least one player must pay for hosting and inference in order to generate revenue through model usage. This implies that total operation costs depend on the final performance achieved after fine-tuning, rather than baseline performance, and the generalist must bear some of the operation costs associated with improved performance. As a robustness check, we examine the effect of changing $G$'s operation cost from $c_\omega \alpha_1 (1 - \omega)$ to $c_\omega \alpha_0 (1 - \omega)$, meaning the generalist does not bear any additional operation costs resulting from downstream model improvements. The coefficients in $G$'s best response change from:

- $A = -\frac{3}{2}(\epsilon - \delta + c_\omega)(\delta + c_\omega)$ to $A = -\frac{(\delta + c_\omega)(\epsilon - \delta)}{2}$,
- $B = (\epsilon - \delta + c_\omega)(1 - \delta) + (c_\omega - \delta)(\delta + c_\omega)$ to $B = \frac{(\epsilon - \delta)(1 - \delta)}{2}$,
- $C = \alpha_0(\epsilon - \delta - 1 + c_\omega) + \frac{(\epsilon - c_\omega)(1 - \delta)}{2}$ to $C = \alpha_0(\epsilon - \delta - 1 + c_\omega)$.

While $U_G$ increases slightly, there are no qualitative differences in the equilibrium outcomes with $c_\omega \in \{0.01, 0.1, 0.5, 1\}$ between the version with $G$'s operation costs as $c_\omega \alpha_0 (1 - \omega)$ vs. $c_\omega \alpha_1 (1 - \omega)$. We add plots for the replication of equilibrium outcomes with the same parameters as Figures 6 and 7 with $\alpha_0$ in Figures 12 and 13.

These two setups are qualitatively indistinguishable because for models with downloadable weights, $D$ will incur most of the operation costs for fine-tuning when $\omega$ is high enough. If $\omega$ is low and the model is mostly closed, the operation costs are divided in one of two ways. First, the closed model provider offers fine-tuning services, but this fine-tuning occurs on the generalist's infrastructure (e.g., OpenAI's fine-tuning APIs). Second, $D$ covers operation costs related to fine-tuning but must choose $\alpha_1 = \alpha_0$ because production costs ($(\alpha_1 - \alpha_0)^2 \omega^{-1}$) become prohibitive if $D$ tries to improve the model when openness is extremely low. In this case, the distinction between $\alpha_1$ vs. $\alpha_0$ in $G$'s operation costs becomes irrelevant because $\alpha_1 = \alpha_0$.

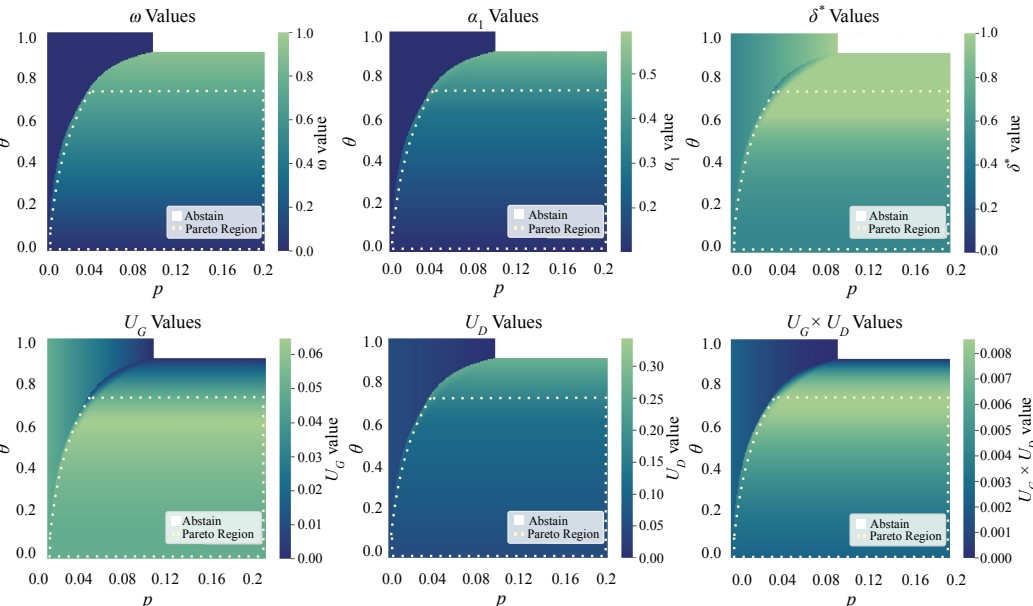

Figure 12: **(Alternate Operation Costs)** When $G$'s operation costs are a function of $\alpha_0$ instead of final performance, equilibrium outcomes for low performance ($\alpha_0 = 0.1, c_\omega = 0.01, \epsilon = 0.1$) with Nash bargaining still create a region where raising the open-source threshold improves $\omega$, $\alpha_1$, and $U_D$ but lowers $U_G$.

The results hold regardless of which player bears the additional operation costs associated with the fine-tuned model's improved performance. In practice, generalists can batch operation costs and take advantage of cost amortization across larger deployment volumes [12], but because our interest is in cost parameters the regulator can directly control, we defer the exploration of hardware and scale efficiencies to future research.

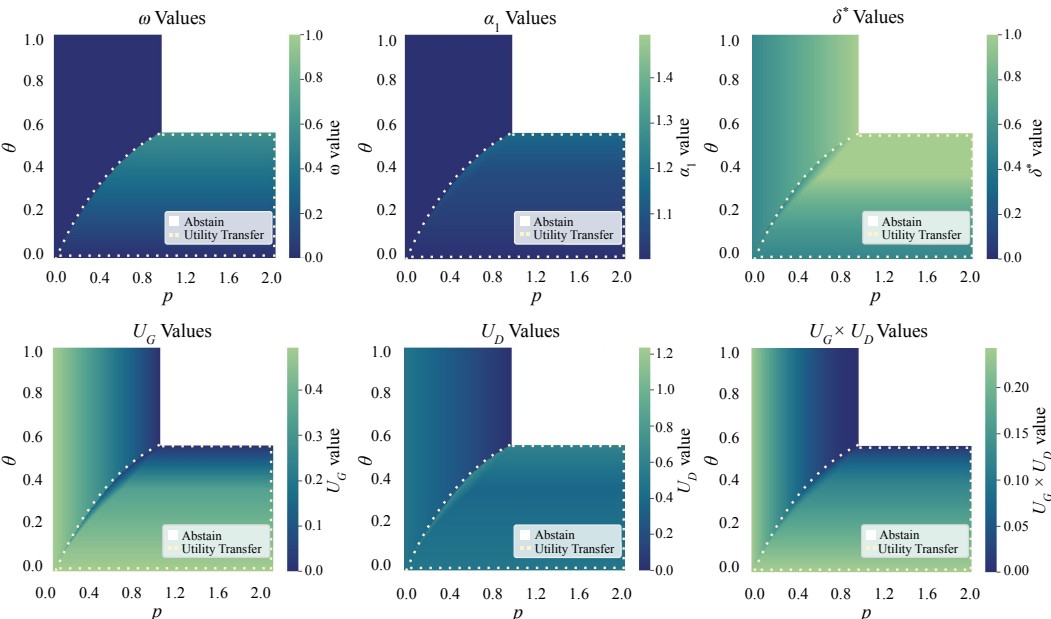

Figure 13: **(Alternate Operation Costs)** When $G$'s operation costs are a function of $\alpha_0$ instead of final performance, equilibrium outcomes for high performance ($\alpha_0 = 1, c_\omega = 0.01, \epsilon = 0.1$) with Nash bargaining show Pareto improvement over utilities $(\omega, \alpha_1, U_G, U_D)$ for any $(p, \theta)$ regulation in the dotted region.

# D  Model Extensions

## D.1  Flexible Initial Performance

$G$'s costs of developing the model to base performance $\alpha_0$ are dropped from $\phi_G$ in our game, as we assume $G$ has already invested in achieving a fixed performance level prior to the regulation and cannot adaptively recover sunk development costs. "Development" is a general label for all fixed costs related to pre-training (data licensing, labor, hardware, R&D) and post-training (alignment, evaluation). This setup resembles the decisions of model developers who have reached a performance ceiling and cannot overcome resource limitations in the short term.

Although the main results assume that $G$'s choice of $\alpha_0$ occurs prior to the introduction of the regulation, an alternative setup may allow $G$ to simultaneously choose $\omega$ and $\alpha_0$ rather than selecting an openness level given a fixed performance level. This setup represents scenarios where new entrants jointly choose performance and openness levels for a model or when incumbents decide to release several variants of a pre-existing model.

**Proposition 5.** *When $|\epsilon - 1 - \delta| \geq c_\omega$, the best response $(\alpha_0{}^*, \omega^*)$ obtained by calculating $\omega^*$ and finding $\alpha_0^*$ numerically using $\omega^*$ achieves a global maximum for $U_G$.*

*Proof.* $G$'s strategy is two-dimensional:

$$\begin{bmatrix} \alpha_0{}^* \\ \omega^* \end{bmatrix} = \underset{\alpha_0, \omega}{\mathrm{argmax}} \quad U_G$$

From Proposition 1, $U_G$ can be written in terms of $\omega$ and $\alpha_0$.

$$U_G = (\epsilon\omega + \delta(1 - \omega))\left(\alpha_0 + \frac{\omega(1 - \omega(1 - \omega) - c_\omega\omega)}{2}\right) - $$
$$\alpha_0\omega + c_\omega\left(\alpha_0 + \frac{\omega(1 - \omega(1 - \omega) - c_\omega\omega)}{2}\right)(1 - \omega) - p\mathbb{1}[\omega < \theta].$$

Differentiating $U_G$ with respect to $G$'s decision variables, we get

$$\frac{\partial U_G}{\partial \alpha_0} = \epsilon\omega + \delta(1-\omega) - \omega + c_\omega(1-\omega),$$

$$\frac{\partial U_G}{\partial \omega} = (\epsilon - \delta)(\frac{\omega - \delta\omega - (\delta + c_\omega)\omega^2}{2} + \alpha_0) + ((\epsilon - \delta)\omega + \delta)(\frac{1-\delta}{2} - (\delta + c_\omega)\omega) - \alpha_0 +$$

$$c_\omega(\frac{\omega - \delta\omega - \delta\omega^2 - c_\omega\omega^2}{2} + \alpha_0) - (c_\omega - c_\omega\omega)(\frac{1-\delta}{2} - (\delta + c_\omega)\omega)$$

$$\frac{\partial U_G}{\partial \alpha_0\omega} = \epsilon - \delta - 1 - c_\omega,$$

$$\frac{\partial U_G}{\partial \omega\alpha_0} = \epsilon - \delta - 1 + c_\omega,$$

$$\frac{\partial^2 U_G}{\partial \alpha_0^2} = 0.$$

From the derivation in the proof of Proposition 2, we know that $\frac{\partial U_G}{\partial \omega} = A\omega^2 + B\omega + C$, where $A = -\frac{3}{2}(\epsilon - \delta + c_\omega)(\delta + c_\omega)$, $B = (\epsilon - \delta + c_\omega)(1-\delta) + (c_\omega - \delta)(\delta + c_\omega)$, and $C = (\epsilon - \delta - 1 + c_\omega)\alpha_0 + \frac{(\delta - c_\omega)(1-\delta)}{2}$. Therefore,

$$\frac{\partial^2 U_G}{\partial \omega^2} = 2A\omega + B$$
$$= -3(\epsilon - \delta + c_\omega)(\delta + c_\omega)\omega + (\epsilon - \delta + c_\omega)(1-\delta) + (c_\omega - \delta)(\delta + c_\omega)$$
$$= (\epsilon - \delta + c_\omega)(1 - \delta - 3(\delta + c_\omega)\omega) + (c_\omega - \delta)(\delta + c_\omega).$$

$$\nabla^2 U_G = \begin{bmatrix} \frac{\partial^2 U_G}{\partial \alpha_0^2} & \frac{\partial^2 U_G}{\partial \alpha_0 \partial \omega} \\ \frac{\partial^2 U_G}{\partial \omega \partial \alpha_0} & \frac{\partial^2 U_G}{\partial \omega^2} \end{bmatrix}$$
$$= \begin{bmatrix} 0 & \epsilon - 1 - \delta - c_\omega \\ \epsilon - \delta - 1 + c_\omega & (\epsilon - \delta + c_\omega)(1 - \delta - 3(\delta + c_\omega)\omega) + (c_\omega - \delta)(\delta + c_\omega). \end{bmatrix}$$

Next, we show that the Hessian of $U_G$ is negative semidefinite when $|\epsilon - 1 - \delta| \geq c_\omega$. By inspection, the first principal minor satisfies $\frac{\partial^2 U_G}{\partial \alpha_0^2} \leq 0$. The criterion $\det(\nabla^2 U_G) \geq 0$ only holds when $|\epsilon - 1 - \delta| \geq c_\omega$.

$$\det(\nabla^2 U_G) = 0 - (\epsilon - 1 - \delta - c_\omega)(\epsilon - \delta - 1 + c_\omega)$$
$$= -(\epsilon - 1 - \delta)^2 + c_\omega^2.$$

Taking the square root of both sides, we get that $\det(\nabla^2 U_G) \geq 0$ only when $|\epsilon - 1 - \delta| \geq c_\omega$, given that $c_\omega \geq 0$ since it is a nonnegative cost ratio. This implies that the Hessian matrix $\nabla^2 U_G$ is negative semidefinite when $|\epsilon - 1 - \delta| \geq c_\omega$, which establishes the concavity of $U_G$ w.r.t. $\alpha_0$ and $\omega$ and guarantees the joint solution achieves a global maximum. $\square$

When the condition $|\epsilon - 1 - \delta| \geq c_\omega$ is not met, $G$ generates almost no positive utility from releasing the model (Figures 14 and 15).

## D.2 Multidimensional Performance Definition

The utility functions assume performance is a one-dimensional property which permits a linear ranking of models. However, it is plausible that no single model consistently dominates across all dimensions (reasoning, general tasks, efficiency). Performance can be extended to a multidimensional variable to capture such scenarios. Formally, this requires the closed-form derivations for both players' utility functions to be reformulated to handle vector-valued performance metrics. For example, $\frac{\partial U_D}{\partial \alpha_1}$

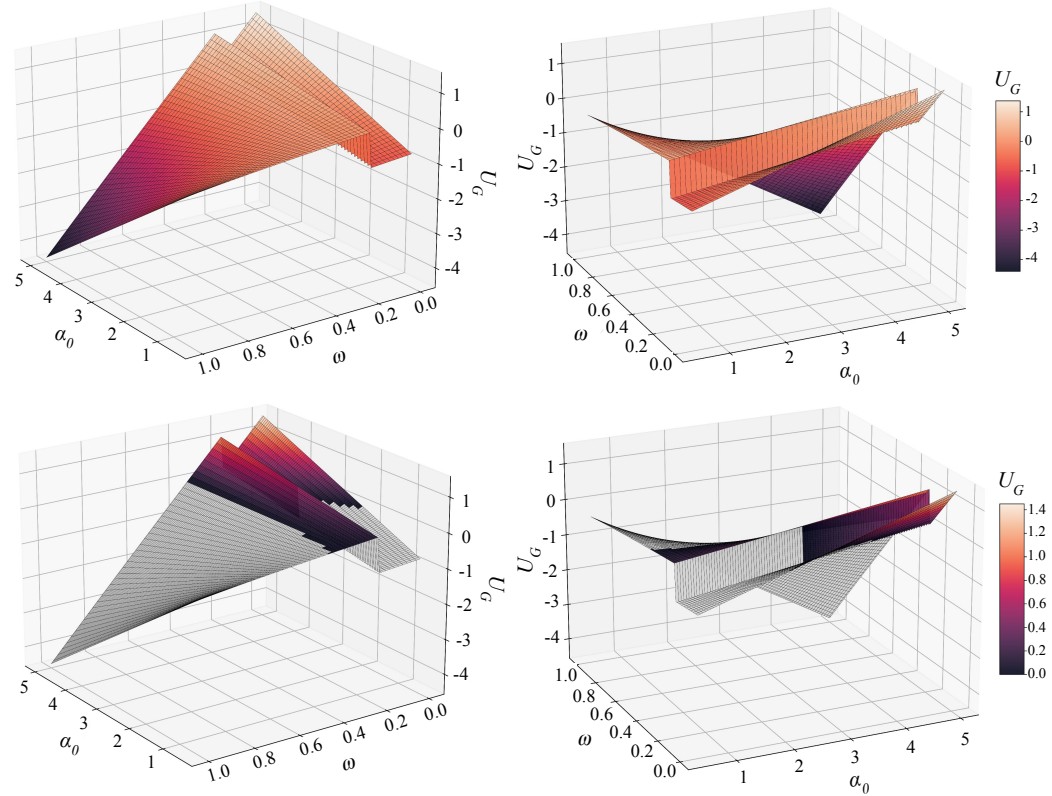

Figure 14: Generalist's strategy space for $(\alpha_0, \omega)$ when $|\epsilon - 1 - \delta| \geq c_\omega$, with parameters $\epsilon = 0.1, \delta = 0.5, c_\omega = 0.01, p = 1, \theta = 0.2$. **(Top)** $G$'s unconstrained strategy space for `azimuth` $= 145$ and `azimuth` $= 245$. **(Bottom)** $G$'s strategy space when abstaining for negative utility payoffs for `azimuth` $= 145$ and `azimuth` $= 245$.

would become an $n$-dimensional vector $\nabla_{\alpha_1} U_D$, where each partial derivative entry represents the marginal utility with respect to a different performance dimension. This setup can encode how regulatory performance preferences diverge from the utility considerations of end users, but because this complexity distracts from the core analysis on model openness, we omit such fine-grained performance definitions from the main results.

## E    Regulation Equilibrium Results

*Proof of Proposition 3.* The area above the indifference curve in the space of $(p, \theta)$ choices represents when $\omega^* \to 0$, while the area below it represents when $G$ meets the open-source threshold.

$$U_G(\omega \approx 0) \geq U_G(\omega = \theta)$$
$$\delta\alpha_1 - c_\omega\alpha_1 - p^* \geq (\epsilon\theta + \delta(1-\theta))\alpha_1 - \alpha_0\theta - c_\omega\alpha_1(1-\theta)$$
$$\delta - c_\omega - \frac{p^*}{\alpha_1} \geq \epsilon\theta + \delta - \delta\theta - \frac{\alpha_0\theta}{\alpha_1} - c_\omega + c_\omega\theta$$
$$-\frac{p^*}{\alpha_1} \geq \theta(\epsilon - \delta - \frac{\alpha_0}{\alpha_1} + c_\omega)$$
$$p^* \leq \theta(\delta + \frac{\alpha_0}{\alpha_1} - \epsilon - c_\omega)\alpha_1$$

For a given threshold, as long as $p \leq p^*$, $G$ will keep the model closed rather than meeting the threshold. If $p^* < 0$, $G$ will always open the model at or above the threshold, regardless of the threshold. $\qquad\square$

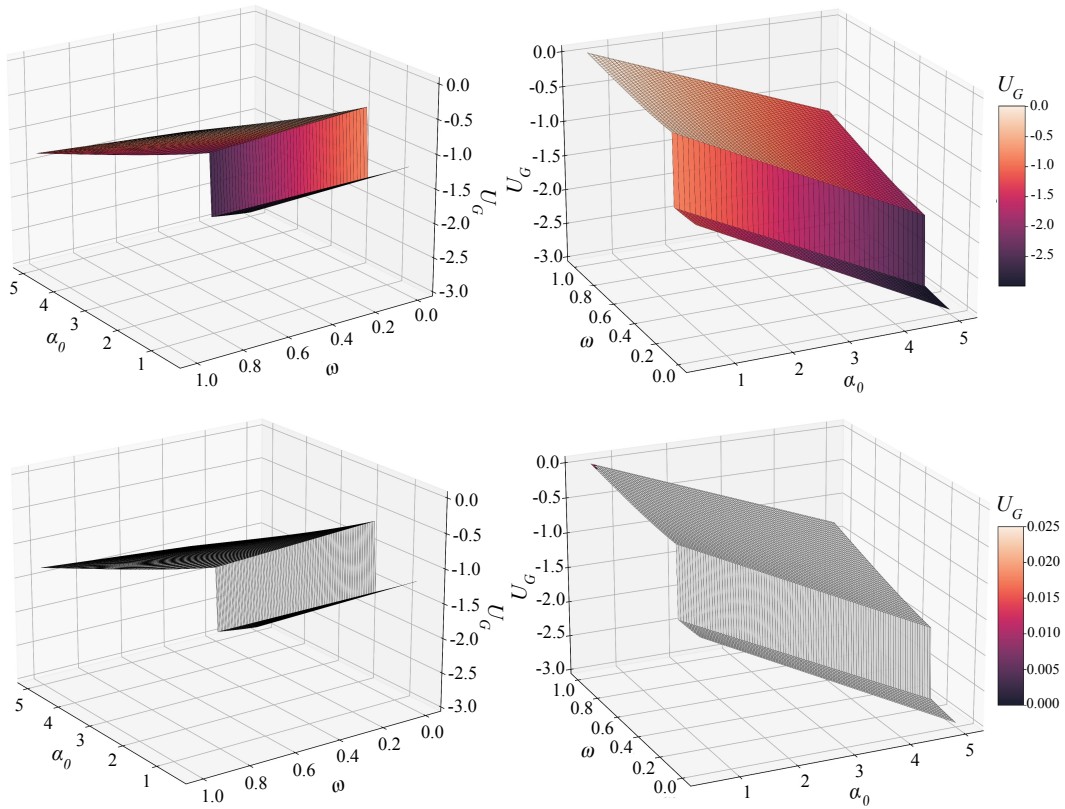

Figure 15: Generalist's strategy space for $(\alpha_0, \omega)$ when $|\epsilon - 1 - \delta| < c_\omega$, with parameters $\epsilon = 0.7, \delta = 0.1, c_\omega = 0.5, p = 1, \theta = 0.2$. **(Top)** $G$'s unconstrained strategy space for `azimuth = 145` and `azimuth = 245`. **(Bottom)** $G$'s strategy space when abstaining for negative utility payoffs for `azimuth = 145` and `azimuth = 245`.

**Proposition 6.** *There exist games with quadratic costs and a monotonic revenue function $r(\alpha_1)$ in which there are Pareto-optimal regulations that improve all utilities $(\omega, \alpha_1, U_G, U_D)$ compared to no regulation.*

*Proof.* Given a game with Nash bargaining and the parameters $\alpha_0 = 0.1, c_\omega = 0.05, \epsilon = 0.1$, the no-regulation equilibrium is $(\omega^* \approx 0, \delta^* = 0.53, \alpha_1^* = 0.1024)$ with $U_D = 0.0480$ and $U_G = 0.0478$. Introducing a regulation of $\theta = 0.6$ and $p = 0.05$ leads to the equilibrium outcome $(\omega^* = 0.6, \delta^* = 0.97, \alpha_1 = 0.2746, U_G = 0.0575, U_D = 0.1090)$ because $U_G(\omega^* = 0.6) > U_G(\omega^* \approx 0)$. This Pareto-dominates no regulation, as $(\omega, \alpha_1, U_G, U_D)$ are all strictly better. Figure 6 visualizes the utility implications over the space of possible regulation tuples $(p, \theta)$. $\qquad\square$

**Proposition 7** (Bounds for Pareto-Optimal Regulation). *Let $\mu > 0$ be an arbitrarily small constant. Given that there exist Pareto-improving regulations for utilities $(\omega, \alpha_1, U_G, U_D)$, the Pareto-optimal region is characterized by upper bounds of $\theta = \frac{1-\delta}{c_\omega - \delta}$, $p_{max}$ and lower bounds of $\theta = \mu$, $p = \theta(\delta + \frac{\alpha_0}{\alpha_1} - \epsilon - c_\omega)\alpha_1 + \mu$.*

*Proof.* To improve $\omega$ beyond the no-regulation equilibrium, any Pareto-optimal regulation must create a binding constraint where $G$ complies with the threshold requirement. There are three cases of the game: (1) $G$ already clears the required openness threshold without regulation, (2) $G$ violates the threshold requirement and does not change strategy, or (3) $G$ adjusts the openness level to the threshold to comply with the regulation. Case (2) is dominated by (1), so the game must fall in (1) or (3). The regulation only has an effect in case (3), implying that the Pareto improvement region must lie under the indifference curve.

By Proposition 3, the lower bound on penalty for the region is given by $p = \theta(\delta + \frac{\alpha_0}{\alpha_1} - \epsilon - c_\omega)\alpha_1 + \mu$, where $\mu > 0$ is a small constant. The upper bound on penalty is $p_{\max}$ because raising the penalty at a given threshold under the indifference curve does not change $G$'s strategy. By Proposition 1, the constraint $\omega \leq \frac{1-\delta}{c_\omega - \delta}$ must be satisfied if $\alpha_1$ is improved. These two constraints imply $\theta \leq \frac{1-\delta}{c_\omega - \delta}$, giving an upper bound on $\theta$. Further, the region of Pareto improvement is lower-bounded by $\theta = \mu$ since any effective regulation must improve $\omega$ through case (3). $\qquad\square$

## F    Data for Figure 3

Although ELO rankings and MMLU scores are used as heuristics to compare model performance, they are not meant to be a comprehensive measure of overall model capability.

**ELO Score** is the model's arena score from the Chatbot Arena LLM Leaderboard. **% Closed** is calculated the same as Figure 4b from Eiras et al. [14], using Table 3's categorization of model components. The percentage is the the portion of components that received a fully closed classification (C1/D1) out of the number of components where such a score is available (excluding ? and N/A labels).

| Model | ELO Score | % Closed |
|---|---|---|
| Pythia | 893 | 0 |
| T5 | 868 | 0 |
| Stable LM | 840 | 0 |
| Llama | 799 | 0.43 |
| DBRX | 1103 | 0.5 |
| Mistral-7B | 1075 | 0.71 |
| Falcon | 1034 | 0.67 |
| PaLM | 1004 | 1.0 |
| Llama 2 | 1093 | 0.8 |
| Command R | 1180 | 0.8 |
| Command R+ | 1215 | 0.8 |
| Llama 3 | 1269 | 0.8 |
| GPT-4 | 1256 | 0.82 |
| Gemini | 1380 | 0.92 |
| Claude 2 | 1132 | 0.92 |

**MMLU Score** uses exact match accuracy for MMLU All Subjects from the HELM leaderboard. MMLU scores marked with (*) are self-reported in the model documentation, rather than collected from HELM evaluations, so the comparisons may be inconsistent.

| Model | MMLU Score | Weights | Release Date |
|---|---|---|---|
| Claude 3.5 Sonnet | 0.873 | Closed | 06-2024 |
| DeepSeek v3 | 0.872 | Open | 12-2024 |
| Gemini 1.5 Pro | 0.869 | Closed | 02-2024 |
| Claude 3 Opus | 0.846 | Closed | 03-2024 |
| Llama 3.1 Instruct Turbo (405 B) | 0.845 | Open | 07-2024 |
| GPT-4o | 0.843 | Closed | 05-2024 |
| Qwen2.5 Instruct Turbo (72B) | 0.834 | Closed | 04-2024 |
| GPT-4 | 0.824 | Closed | 03-2023 |
| Amazon Nova Pro | 0.82 | Closed | 12-2024 |
| GPT-4 Turbo | 0.813 | Closed | 11-2023 |
| Llama 3.2 Vision Instruct Turbo (90B) | 0.803 | Open | 09-2024 |
| Llama 3.1 Instruct Turbo (70B) | 0.801 | Open | 07-2024 |
| Mistral Large 2 | 0.8 | Open | 07-2024 |
| Gemini 2.0 Flash | 0.797 | Closed | 12-2024 |
| Llama 3 (70B) | 0.793 | Open | 04-2024 |
| Llama 3.3 Instruct Turbo (70B) | 0.791 | Open | 12-2024 |
| PaLM-2 (Unicorn) | 0.786 | Closed | 05-2023 |
| Jamba 1.5 Large | 0.782 | Open | 08-2024 |
| Mixtral (8x22B) | 0.778 | Open | 04-2024 |
| Phi-3 (14B) | 0.775 | Open | 05-2024 |
| Qwen1.5 (72B) | 0.774 | Closed | 02-2024 |
| Yi (34B) | 0.762 | Open | 11-2023 |
| Gemma 2 (27B) | 0.757 | Closed | 06-2024 |
| Claude 3.5 Haiku | 0.743 | Closed | 11-2024 |
| DBRX Instruct | 0.741 | Open | 03-2024 |
| Gemini 1.5 Flash | 0.739 | Closed | 05-2024 |
| DeepSeek LLM Chat (67B) | 0.725 | Open | 11-2023 |
| Command R Plus | 0.694 | Open | 08-2024 |
| PaLM-2 (Bison) | 0.692 | Closed | 11-2023 |
| GPT-3.5 Turbo | 0.689 | Closed | 03-2023 |
| Llama 3 (8B) | 0.668 | Open | 04-2023 |
| OLMo 1.7 (7B) | 0.538 | Open | 07-2024 |
| Chinchilla (70B) | *0.676 | Closed | 03-2022 |
| PaLM (540B) | *0.693 | Closed | 04-2022 |
| U-PaLM | *0.707 | Closed | 10-2022 |
| Flan-PaLM | *0.752 | Closed | 10-2022 |
| Llama (65B) | *0.634 | Open | 02-2023 |
| Llama 2 (70B) | *0.689 | Open | 07-2023 |
| Falcon (180B) | *0.706 | Open | 09-2023 |

# G    Additional Figures

The equilibrium of the game is characterized by a triple $(\omega, \delta, \alpha_1)$. Here, we check the effects of varying $\alpha_0, \epsilon$, and $c_\omega \in \{0, 0.01, 0.1, 0.5\}$ on the no-regulation equilibrium, $(\omega, \delta, \alpha_1)$ for $p = 0$, under different bargaining solutions. Figure 2 uses $c_\omega = 0.1$ over different $\alpha_0$ and $\epsilon$ values.

## G.1    Openness Levels

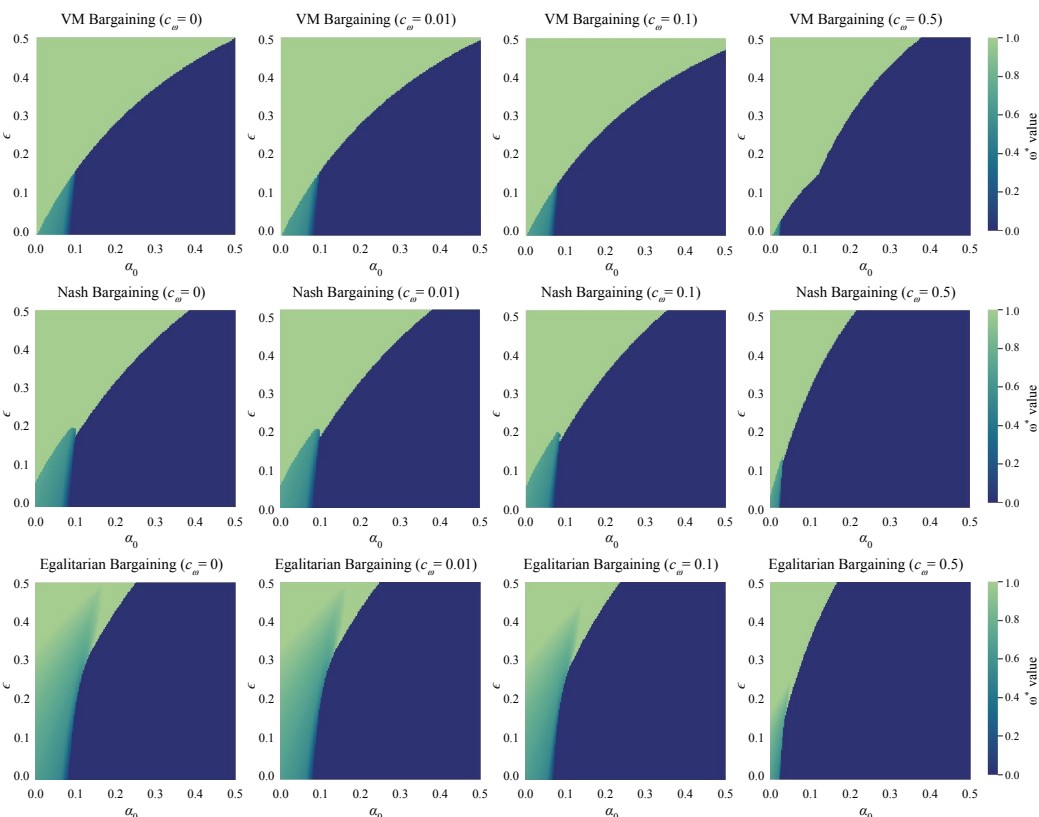

Figure 16: Openness levels $(\omega^*)$ chosen by $G$ without regulation $(p = 0)$ under VM, Nash, and egalitarian bargaining solutions. Rows correspond to bargaining solutions, and operation costs increase from left to right: $c_\omega = 0, 0.01, 0.1, 0.5$. Increasing $c_\omega$ causes less intermediate values of openness in the region of low $\alpha_0, \epsilon$ because $G$ avoids operation costs by fully opening. The fully open area at high $\alpha_0, \epsilon$ recedes as $c_\omega$ increases for Nash and egalitarian bargaining solutions because $D$'s utility is harmed by high operation costs for high performance when $G$ receives the majority of revenue through $\delta^*$.

## G.2 Bargaining Coefficients

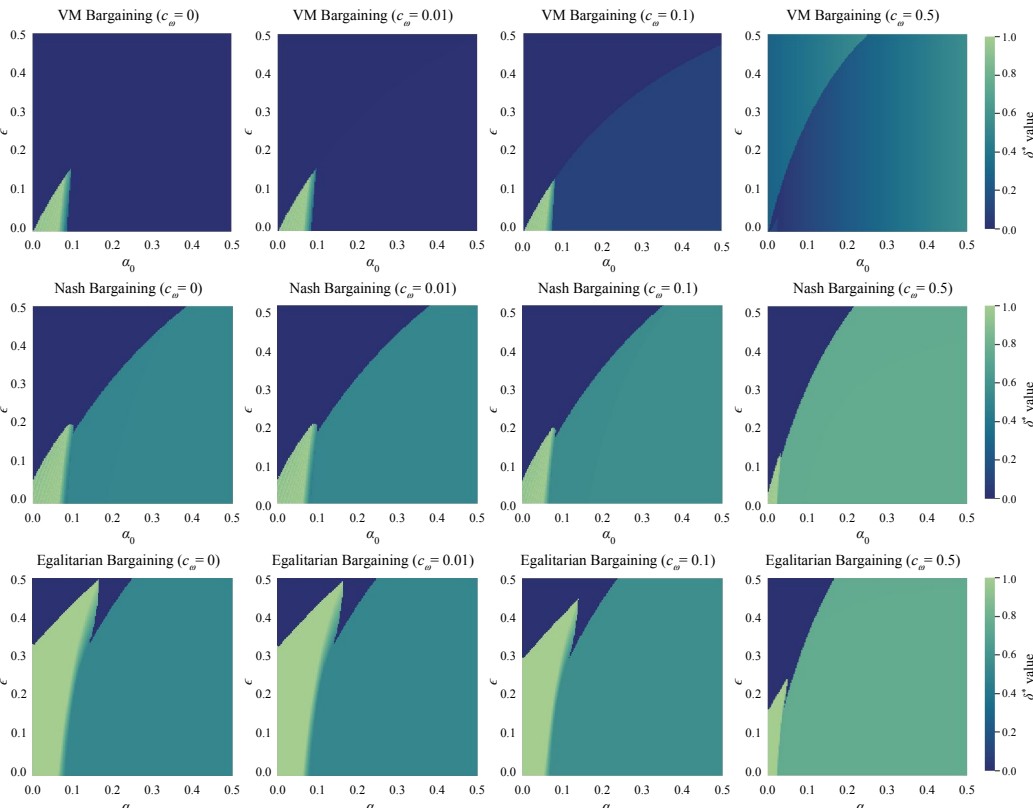

Figure 17: Bargaining coefficients ($\delta^*$) jointly chosen by $G$ and $D$ without regulation ($p = 0$) under VM, Nash, and egalitarian bargaining solutions. Rows correspond to bargaining solutions, and operation costs increase from left to right: $c_\omega = 0, 0.01, 0.1, 0.5$. For all three bargaining solutions, higher operation costs push $\delta$ to favor $G$ more in regions where $G$ fully closes the model. For low $\alpha_0$, where $G$ fully opens the model and there is less revenue, the bargain favors $D$ more.

## G.3 Final Performance

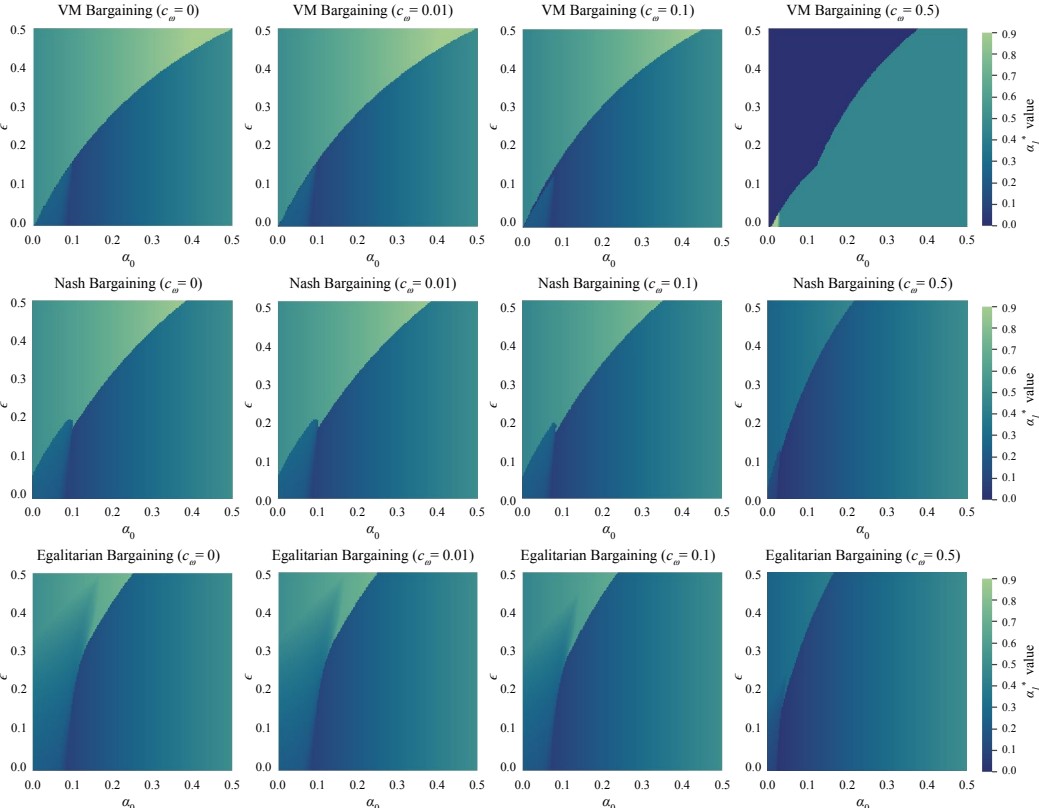

Figure 18: Improvement level ($\alpha_1^*$) chosen by $D$ without regulation ($p = 0$) under VM, Nash, and egalitarian bargaining solutions. Rows correspond to bargaining solutions, and operation costs increase from left to right: $c_\omega = 0, 0.01, 0.1, 0.5$. $D$'s level of improvement decreases as operation costs increase across all bargaining solutions, and regardless of whether $G$ fully opens or closes the model.

