# OpenReview forum: "Modeling the Economic Impacts of AI Openness Regulation"
_NeurIPS.cc/2025/Conference — NeurIPS 2025 poster_

### Official Review · Reviewer_fUP1 · 2025-06-06

**Clarity:** 3
**Significance:** 2
**Originality:** 2
**Rating:** 5
**Confidence:** 4

**Summary:**

In an effort to understand and predict the impacts of AI governance policy in relation to open-source general-purpose AI foundation models, the authors introduce a game-theoretic model for assessing and predicting the effects of AI governance regulation thresholds and penalties on decisions made by the producers and refiners of open-source general-purpose AI models. The approach models the effects of strategic decisions made by AI developers to adjust levels of openness along a spectrum rather than under an open or closed binary construct. To reveal possible open-source value chain dynamics, the model allows for various levels of openness to be set by the producer of the general-purpose model, and weighs impacts of other criteria and forces such as revenue-related bargaining decisions between generalists and specialists, and regulatory open-source thresholds.

**Questions:**

•	Did the authors consider factoring-in the distinctions among how G, D, and regulators define model improvements or performance? As noted in the Weaknesses section, the regulatory goals addressed in the modeling should be better articulated.
•	Did the authors consider what group or groups are their target audience? It might be helpful for the reader to understand this to get a better sense of the ultimate goals for application of the model.

**Ethical Concerns:**

["NO or VERY MINOR ethics concerns only"]

**Final Justification:**

Following positive engagement with the authors which improved the paper, I've increased the rating from borderline accept to accept.

**Limitations:**

The authors acknowledge some limitations and caveats to their proposed framework; for example, their stated limitations acknowledge some important areas for future research and model improvement, in particular inclusion of “social utility” related criteria in the modeling.

**Paper Formatting Concerns:**

•	There is a typo in line 294: “The effectiveness of openness regulation is highly sensitive to the TEH generalist’s initial model…”

**Quality:**

3

**Strengths And Weaknesses:**

Strengths
•	The game is valuable as a way to understand how some regulatory approaches might affect AI developers, whether regulatory approaches could work, and areas where they might fail.
•	The authors advance prior work in a way that somewhat reflects today’s regulatory and AI business landscapes, including by modeling for some of the nuances and complexities present in real-world scenarios.
•	The authors are clear in their explanation of the game’s role players, forces and factors affecting predicted outcomes, and how the game works.

Weaknesses
•	An overarching and relatively obvious weakness of this work is the game’s existence as a theoretical exercise that -- despite its addition of nuances prior work does not consider (such as incorporating impacts of openness along a spectrum rather than an open/closed binary) – does not consider several other real world factors that could limit the model’s predictive capabilities.
•	For instance, as just one example, the model does not consider the impacts of venture capital funding which could allow generalists or specialists to tolerate lower revenue goals. It also does not consider political forces, or other factors that might disrupt the model.
•	The authors do not adequately define the regulatory goals their model addresses or considers. It is somewhat implied that the goals the model is designed to address are “innovation,” model performance as measured by industry-designed benchmarks, and possibly competition related goals, rather than other regulatory goals that are also facilitated through open-sourcing such as model transparency and explainability related goals. But it is important to better define these regulatory goals.
•	The paper does not adequately define or explain reputational benefits, how they are measured, or how the size or market power of a company might affect reputational benefits/premiums.
•	A minor point, but the authors could more directly explain and cite the Nash bargaining solution.

---

> ### Author Rebuttal · Authors · 2025-07-30
>
> Thanks for the thoughtful engagement with our work. We appreciate your positive feedback and address your questions and suggestions below.
>
> # Regulatory Goals
>
> >  It is somewhat implied that the goals the model is designed to address are “innovation,” model performance as measured by industry-designed benchmarks, and possibly competition related goals, rather than other regulatory goals that are also facilitated through open-sourcing such as model transparency and explainability related goals.
>
> *Defining Regulatory Goals*.
>
> We agree that the regulatory goals should be stated more explicitly. To handle this, we’ve moved the formalism of a Pareto-optimal policy (Def. 1) from the Appendix into 4.2. The regulator can choose any set of utility functions (i.e., $\alpha_1, \omega, U_D, U_G$) and any weighting of those utility functions to define regulatory objectives. At the same time, we avoid committing to a particular balance of objectives, since we want to illustrate the full space of possible regulatory approaches.
>
> While the regulator could be modeled as a third player with an explicit utility function, we intentionally refrain from defining specific regulatory preferences. This enables flexible parameterization of the model according to different objectives, rather than constraining analysis to a particular social utility function that may not suit diverse regulatory contexts. We also want to note that providing a more granular definition of regulatory utility beyond what Def. 1 offers would invite the well-documented complexity of ranking and weighting various objectives from social choice and welfare aggregation literature.
>
> *Other Open-Sourcing Goals.*
>
> Our analysis assumes $\omega$ is a 1D variable that captures a property like model access. One could imagine using an equivalent model for reasoning about a spectrum of transparency or explainability measures. Sec. 5.2 now mentions that model transparency and explainability can be captured *in addition to* model access in future work: “While our analysis assumes that higher model openness and performance are generally beneficial, a regulator must consider safety risks associated with higher performance or market-wide loss scenarios, such as possible intermediate levels of openness where consumers may be worse off. This complexity suggests that defining openness as a multidimensional property may be valuable to capture factors beyond simple model access, such as documentation and explainability of model outputs.”
>
> # Modeling Assumptions
>
>
> > The paper does not adequately define or explain reputational benefits, how they are measured, or how the size or market power of a company might affect reputational benefits/premiums.
>
> We roughly capture the issue of market power in $G$’s utility function by scaling the reputational premium by model performance. The $\epsilon \alpha_1$ term in $U_G$ means firms with superior models receive greater reputational benefits from openness.
>
> To better explain reputational benefits, Sec. 2.2 now references extensive prior work on defining reputational benefits in the context of OSS contributions and commons-based production:
>
> * “Coase’s Penguin, or, Linux and The Nature of the Firm” (2002), which discusses reputation as an “indirect appropriation mechanism.”
> * “Why develop open-source software? The role of non-pecuniary benefits, monetary rewards, and open-source licence type” (2007), which profiles OSS developer motivations.
> * “Social Status in an Open-Source Community” (2005), which has a case study on OSS reputation incentives.
>
> In our case, $\epsilon$ captures reputational advantages within developer and research communities, which may aid in recruitment and third-party integration with the product. We don’t provide an exact approach for measuring $\epsilon$, though it should in general be low (at least $< 0.5$), given that reputational benefits generally do not exceed the amount of closed revenue that can be gained from monetizing a model.
>
> > as just one example, the model does not consider the impacts of venture capital funding which could allow generalists or specialists to tolerate lower revenue goals.
>
> This is an interesting point. We have updated L102 to explicitly state the assumption that players abstain when their utility is negative. Venture capital might lower revenue goals because it provides capital that firms can sit on, but on the other hand, it tends to push companies towards high risk and fast growth. Modeling scenarios where players tolerate lower revenues in expectation of future profits requires introducing time variance and additional parameters specifying players’ tolerance for short-term losses. This would substantially increase model complexity and require additional assumptions about discount rates and availability of capital. We have therefore added modeling time variance and multi-firm competition as an important future direction in Sec. 5.2.
>
> > It also does not consider political forces, or other factors that might disrupt the model.
>
> Social utility considerations of the regulator give some capacity for modeling political factors -- e.g., political pressures associated with setting the penalties and thresholds too high. We allude to this in L255 when discussing “wasted enforcement costs” associated with ineffective, high thresholds. However, this could be formalized alongside Def. 1 by applying a regulator penalty for different $(\theta, p)$ combinations:  ${\arg\max}_{p, \theta}\  w^\top \mathbf{u} - \lambda \theta p \  \quad s.t.\ \|w\|_1=1, w_i>0, \forall i \in [d]$.
>
> Our paper focuses on levers that the regulator can directly control (see Appx. C for our justifications for omitting considerations like deployment scale). We abstract away from exogenous factors to isolate causal relationships between policy instruments and equilibrium outcomes. However, we acknowledge that several factors may disrupt the model’s outcomes in practice, and future work can systematically characterize these factors or add stochasticity to examine how the model performs under uncertainty or unexpected market changes.
>
> > An overarching and relatively obvious weakness of this work is the game’s existence as a theoretical exercise that -- despite its addition of nuances prior work does not consider (such as incorporating impacts of openness along a spectrum rather than an open/closed binary) – does not consider several other real world factors
>
> We acknowledge that our work relies on several simplifying assumptions, though these abstractions serve two important purposes. First, this approach follows established practices in economic modeling, where parsimony yields more robust and interpretable insights by isolating the most direct relationships between regulation and developer choices. Second, the abstract definitions of $\omega, \alpha_0$ are designed to provide flexibility over how such measures are constructed, allowing the model to remain general while accounting for relative magnitude. This abstraction is meant to accommodate different measurement capabilities, rather than tying the model to particular technical definitions – i.e., $\alpha_0$ may cover efficiency or latency considerations beyond benchmark accuracy alone.
>
> # Miscellaneous
>
> > Did the authors consider factoring-in the distinctions among how G, D, and regulators define model improvements or performance?
>
> When constructing the model, we did consider how regulators may consider emissions targets or safety as part of performance, while $G$ and $D$ may focus more narrowly on commercial viability or benchmarks. Our model is designed to evaluate regulator-defined objectives, so we maintain $\alpha_0$ as a metric focused on task accuracy and efficiency and rely on the regulator’s utility function / discretion to incorporate broader environmental, safety, or equity concerns that diverge from firms' revenue-maximizing definition of performance. However, we recognize this is an important modeling consideration and have added a discussion in Sec. 5.1 about how regulators might choose different definitions of *average* performance: “Given that models may excel in certain domains while underperforming in others, regulators must determine whether to use composite performance metrics, domain-specific benchmarks, or capability-weighted averages to define model performance, with each approach reflecting different model development priorities.”
>
> > A minor point, but the authors could more directly explain and cite the Nash bargaining solution
>
> We have added a citation on L188 and included a discussion of properties and limitations of the bargaining solutions: “Our analysis mainly examines Nash bargaining, which uniquely satisfies independence of utility origins and units, Pareto efficiency, symmetry, and independence of irrelevant alternatives. The two alternative bargaining solutions we consider (VM and Egalitarian) are also Paretian, symmetric, and independent of utility origins. However, generalists and specialists may arrive at non-standard bargaining agreements in practice which violate these properties, especially when utilities are expressed in currency and players have bounded rationality.”

---

> > ### Comment · Reviewer_fUP1 · 2025-08-01
> >
> > Thanks to the authors for considering these review comments and for their thoughtful responses. I appreciate the additions to the paper based on some of these comments.
> >
> > In regard to the additional references to prior work related to reputational benefits, I suggest the authors seek out more recent work that considers a more current understanding of open sourcing. The references are around 20 years old or more.

---

> > > ### Author Response · Authors · 2025-08-01
> > >
> > > We are glad to hear that you found our rebuttal thoughtful. We can incorporate the following more recent works discussing reputation-driven mechanisms for OSS.
> > > * Fukawa et al., “Dynamic Capability and Open-Source Strategy in the Age of Digital Transformation” (2021): Open-source strategy “enables a firm to enhance the attractiveness of its product as users benefit from larger user, complements, and producer networks.”
> > > * The Comingled Code: Open Source and Economic Development (2010): Software firms offer open-source products to exploit their reputational capital, and developers primarily contribute to projects with open-source licenses because higher reputation gains can be obtained.
> > >
> > > There are limited academic sources for AI open sourcing due its recency, but some industry documents discuss reputational motivations -- e.g., Meta's statements on Llama.
> > > * Meta Platforms, Inc. (META), Fourth Quarter 2023 Results Conference Call: “open source is hugely popular with developers and researchers. We know that people want to work on open systems that will be widely adopted, so this helps us recruit the best people at Meta.”
> > >
> > > If the reviewer is aware of recent peer-reviewed work we may have missed, we would greatly appreciate those suggestions.

---

> > > > ### Comment · Reviewer_fUP1 · 2025-08-06
> > > >
> > > > I think the paper will benefit from those additional citations. The Meta conference call comment is really interesting and helps bolster the concept of reputational benefits from open sourcing.

---

> > > ### Comment · Reviewer_fUP1 · 2025-08-06
> > >
> > > A few more possible citations you might add:
> > > https://www.centeraipolicy.org/work/us-open-source-ai-governance
> > > https://www.linuxfoundation.org/research/economic-impacts-of-open-source-ai
> > > https://www.mckinsey.com/capabilities/quantumblack/our-insights/open-source-technology-in-the-age-of-ai#/

---

### Official Review · Reviewer_Rs1J · 2025-06-29

**Clarity:** 4
**Significance:** 2
**Originality:** 3
**Rating:** 5
**Confidence:** 4

**Summary:**

The paper proposes a two-stage game-theoretic model that links a foundation-model developer (generalist $G$) and a downstream fine-tuner (specialist $D$) who bargain over revenue, while a regulator sets an openness threshold and penalty. The game progresses in multiple stages. First, G chooses an openness level on a 0-1 continuum (Fig. 1) that governs whether the model clears the regulatory threshold and what costs it bears. Then, D decides whether (and how far) to fine-tune the model to a new performance level.
The authors derive closed-form expressions for D’s best response and for G’s sub-game-perfect openness choice. The authors demonstrate three policy regimes: (i) Pareto-improvement when $\alpha_0$ is low and moderate, which push G to open the model, (ii) utility-loss zones where penalties only transfer surplus without added openness, and (iii) utility-transfer zones where tighter thresholds reroute surplus from G to D, raising $\alpha_1$ but lowering $U_G$.

The authors claim this provides guidance for calibrating openness rules in the EU AI Act and similar regimes.

**Questions:**

Game theoretic analyses are a fantastic way to provide quantification and grounding for policy makers. This paper is commendable in doing exactly that. However, there are a few areas in which the authors can improve their paper.

The model presently assigns the entire regulatory burden to the generalist. Downstream fine-tuners, however, can face disclosure or safety-testing duties under both the EU Act (Article 28) and the US NIST RMF. I would like to see a specialist-side penalty term (or a brief formal argument for why it is negligible). If introducing that term alters the qualitative comparative statics, please highlight where; if it does not, you gain credibility by demonstrating robustness.

The paper assumes performance improvements exhibit diminishing returns as the base model approaches the frontier, and you treat performance as the only strategic dimension. Yet fine-tuning often delivers non-marginal gains on underperforming tasks, and many practitioners optimize for latency or parameter efficiency instead of raw benchmark scores. I ask for (a) at least one empirical learning-curve plot or citation supporting the concavity assumption, and (b) a short discussion of how a second axis such as inference cost would change the results. Demonstrated robustness to alternative tuning curves or an extended two-dimensional game would lift my novelty and insight scores; silence on the issue would pull them down.

**Ethical Concerns:**

["NO or VERY MINOR ethics concerns only"]

**Final Justification:**

Assuming the authors include the clarified results as a result of our discussion into the paper, my concerns are met and I am comfortable recommending an accept!

**Limitations:**

Yes

**Quality:**

2

**Strengths And Weaknesses:**

# Strengths
The paper tackles a policy problem that is both immediate and under-specified: how openness rules in upcoming AI regulation will shape economic incentives along the foundation-model value chain. By allowing the "generalist” developer, rather than the regulator, to choose an openness level endogenously, the authors capture a strategic knob that existing game-theoretic work leaves fixed. Their closed-form solutions make the comparative statics transparent, so readers can see precisely when higher openness lowers total welfare versus when it re-allocates surplus between the upstream model builder and downstream fine-tuner. Together, these features turn what could have been an abstract toy model into a practically useful lens for policymakers and practitioners who must weigh documentation burdens, revenue-sharing, and compliance costs.

1. The model addresses a live regulatory gap rather than an abstract problem. The EU AI Act exemptions hinge on an ill-defined notion of "open-source", and industry "open-washing” is widely discussed problem.
2. This provides a realistic strategic lever for model developers. The relationship between producers of a general model and those building derived specialist models is under-studied. This extends the relevant game literature by letting G choose openness endogenously. Previous work treats openness as fixed.
3. This work allows clear comparative-statics reasoning instead of pure simulation by obtaining closed-form best responses.

Overall, this work can help readers reason about who bears which costs at different openness levels.

# Weaknesses
Several conceptual and modeling shortcuts undermine that practical promise. The paper treats "foundation models” as uniquely troublesome for open-source definitions, yet earlier software and ML precedents raise identical dilemmas. This framing overstates novelty.

More overall, who decides what is high or low performance? These are vague notions the work depends on, but are never defined. The window of relevant performance changes over time? Once upon a time, GPT-3 was very high performing, but today we would call it low performing. The change in performance could trigger changes in openness. This analysis should minimally define performance, and to improve the modeling, introduce time variance.

1. L24: "Foundation models complicate the standard open-source definition". The complication is not unique to foundation models. Binary-only drivers and source-available databases raised the same debates years ago. It's worth citing "The Value of Open Source Software" by Hoffman et. al. [0]
2. L42-43: The discussion of specialists and fine-tuning appears could be foreshadowed earlier in the introduction. It's worth discussing fine-tuning and the game this implies before this statement.
3. L113: "dividing revenue from foundation model development". The usage of the term "foundation model" does not add much to the discussion. Again, this phenomenon is not unique in the ML world to just foundation models. Look at Gupta et. al. for a discussion on the issues raised by the usage of "foundation" models in policy-related discussions [1]
4. L125-126 and L130: "$\phi_G(\omega, \alpha_0|\theta) = ... + c_{\omega}\alpha_1(1 - \omega) + ...$" Should this be $\alpha_1$ or $\alpha_0$? Currently this implies that operation costs hinge on the final performance reached after fine-tuning, not the baseline. Why does the generalist bear this cost?
5. L125-126: Why are there not regulatory costs for the specialist? Fine-tuned models could be closed-source and/or regulated as well.
6. L127: "there are additional costs to documents and releasing the model publicly which scale with higher levels of openness and performance". Sure, but this is not linearly so? There are already costs associated with documentation internally, as well as the cost of a proper internal model release process. Perhaps this is better modeled as log scaling rather than linear?
7. L128: "safety risk management for open models". This is a regulatory cost, but you are trying to provide an example for production costs.
8. L136: "since incremental improvements are more expensive at higher performance levels..." The entire point of fine-tuning a general model is because it was initially not performing well on the desired downstream task. Can you justify quadratic cost? Empirical fine-tuning curves often show super-linear returns for early steps. Look at literature like Xia et. al. [2]
9. L139: "fine-tune or improve". There are plenty of ways to improve the performance of a closed-source model. In-context learning, RAG, tool use, etc. Maybe just say "fine-tune" instead?
10. L148-149: "model performance is the only feature in $G$'s strategy space". This is a very limiting assumption. Generalists also worry about efficiency. Reference the debate around DeepSeek, o3, o3-mini, and new nano-style models.
11. L157: "higher openness increases production costs". There is ample evidence showing this to be false. Projects like YOLO, vLLM, etc. have benefited drastically from open-souce contributions. Outside of ML, look at Docker, Redis, etc. Even within foundation models, improvements to Gemma have led to improvements in Gemini.

# Checklist concerns
For question 5, the authors say "no" to open access to data and code. But, the authors reference a numerical simulation to produce their figures. Where is the code for that? The justification that "the plots can be easily reproduced" is hand-wavey. Ok, if it is easy, then just share the code?

For question 8, while the simulations can be trivially run on a CPU, a few extra words saying for just how long or how many CPUs would be useful.

# Typos
L221: "effectsacross" -> "effects across"
L222: "identifies conditions the" -> "identifies the conditions for" ?
L294: "teh" -> "the"

[0] Hoffmann, Manuel, Frank Nagle, and Yanuo Zhou. "The Value of Open Source Software." Harvard Business School Working Paper, No. 24-038, January 2024.

[1] Gupta et al. "Data-Centric AI Governance: Addressing the Limitations of Model-Focused Policies." arXiv preprint arXiv:2409.17216 (2024).

[2] Y. Xia et al., "Understanding the Performance and Estimating the Cost of LLM Fine-Tuning," 2024 IEEE International Symposium on Workload Characterization (IISWC), Vancouver, BC, Canada, 2024, pp. 210-223, doi: 10.1109/IISWC63097.2024.00027.

---

> ### Author Rebuttal · Authors · 2025-07-30
>
> We thank the reviewer for engaging carefully with our work. The feedback has helped us refine explanations of our model’s assumptions.
>
> # Robustness
>
> > I would like to see a specialist-side penalty term (or a brief formal argument for why it is negligible).
>
> We derived closed-form results and ran simulations with a specialist-side penalty ($\phi_D$ includes a $p \mathbb{1}[\omega < \theta]$ term). The qualitative takeaways remain the same, since the solutions for ${\alpha_1}^\*$  and $\omega^\*$ do not change, but the  specialist penalty increases $D$’s region of abstaining above the indifference curve because the feasibility condition becomes $\omega \leq \frac{1}{\delta - c_\omega}(\frac{p}{\alpha_0} + \delta - 1)$ instead of $\omega \leq \frac{1-\delta}{c_\omega - \delta}$. There’s also a slight effect on $\delta^*$ and joint utilities above the indifference curve because $U_D$ goes down by $p$, but this does not change our Pareto improvement and utility transfer findings. We have added the results to Appx. B.
>
> > I ask for (a) at least one empirical learning-curve plot or citation supporting the concavity assumption, and (b) a short discussion of how a second axis such as inference cost would change the results. Demonstrated robustness to alternative tuning curves or an extended two-dimensional game would lift my novelty and insight scores; silence on the issue would pull them down.
>
> **(a)** We assume $U_D$ is concave in $\alpha_1$ because higher performance should only benefit $D$ up to a critical point before further investments in fine-tuning cause utility to taper off (costs go up but performance and revenue do not) -- see: Springer et al. “Overtrained Language Models Are Harder to Fine-Tune” (ICML 2025). We don’t claim anywhere that $U_G$ is concave, but we believe most utility functions should be concave because the optimum will always be on a boundary otherwise ($\omega^\*=0, \omega^\*=1$ and ${\alpha_1}^\*=\alpha_0, {\alpha_1}^\* = \infty$). Concavity guarantees $D$ will adopt high-performing models as-is, choosing $\alpha_1 = \alpha_0$ instead of always fine-tuning (addressing your concern with L136).
>
> **(b)** We’ve moved Prop. 3 from Appx. D into the main body of the paper for an analytical way to inspect how something like inference ($c_\omega$) affects openness choices. We’ve also  added discussion of inference costs and a 2D game (jointly optimizing $\omega^\*, {\alpha_0}^\*$) to the Appx.
>
> > Generalists also worry about efficiency. Reference the debate around DeepSeek, o3, o3-mini, and new nano-style models.
>
> "[m]odel performance is the only feature in G's strategy space" describes prior work (the paper which introduced the framework of fine-tuning games). Our paper presents a 2D game with *both* openness and performance in $G$’s strategy space, though the main results analyze when $G$ decides an openness level given $\alpha_0$.
>
> We capture efficiency through the $c_\omega$ constant (L145). The operation cost to performance ratio should be low in the case of high-capability, small models like those you cite but closer to 1 (or even > 1) for inefficient models. Our rationale for not representing efficiency as a strategic dimension is that translating investments into efficiency improvements is dependent on external technological and resource constraints (e.g., hardware innovation, energy capacity) and *model-specific* relationships between architecture choices and inference costs (consistent with the Xia et al. reference you provided). As noted in Appx. C, efficiency is an important dimension for future work, but our paper is primarily concerned with how model openness, as a policy-relevant dimension, affects players’ choices.
>
> # Weaknesses
>
> > The window of relevant performance changes over time? Once upon a time, GPT-3 was very high performing, but today we would call it low performing.
>
> Figure 3(b) is meant to address this concern, as it shows that high vs. low performance is determined relative to the *current* generation of models.
>
> > who decides what is high or low performance?
>
> This is relative (see above), but we’ve added clarification to Sec. 5.1: “Given that models may excel in certain domains while underperforming in others, regulators must determine whether to use composite performance metrics, domain-specific benchmarks, or capability-weighted averages to define model performance, with each approach reflecting different model development priorities.”
>
> > This analysis should minimally define performance, and to improve the modeling, introduce time variance.
>
> Sec. 2.2 has been updated: “Performance represents a coarse measure of the model's accuracy on target tasks, generation speed and efficiency, and user satisfaction.”
>
> We acknowledge that our work relies on several simplifying assumptions. This follows established practices in economic modeling, where parsimony yields more robust and clear insights. While we agree time variance is an important consideration, it substantially complicates the analysis (see response to Reviewer 1) without meaningfully strengthening our core theoretical insights about regulation. That said, we appreciate the modeling suggestions and have noted time variance as a future direction.
>
> > The paper treats "foundation models” as uniquely troublesome for open-source definitions, yet earlier software and ML precedents raise identical dilemmas.
>
> We understand the concern and didn't mean to suggest they’re fully unique (see L607 in Appx.). Our position is that foundation models raise the stakes of open-source definitions because (1) they implicate a broader set of capabilities and downstream uses, (2) achieving reproducibility requires more than open code and data, and (3) high inference costs amplify the advantages of open-sourcing. We have incorporated Gupta et al. as a caveat to our terminology and rewritten Sec. 1 to make this position clearer.
>
> > It's worth citing "The Value of Open Source Software" by Hoffman et. al. [0]
>
> Hoffman et al. was cited in our related works (L50).
>
> # Cost Modeling
>
> Sec. 2.3 has been rewritten to distinguish when we’re referring to the structure of a utility function (broader classes of production, regulatory, and operation costs) vs. a specific functional form (quadratic cost and monotonic revenue).
>
> > There are already costs associated with documentation internally, as well as the cost of a proper internal model release process. Perhaps this is better modeled as log scaling rather than linear?
>
> This is a good point. We ran numerical simulations with log scaling, and the results are qualitatively the same. $G$’s best response is similar to the one derived in Appx. B, except it requires solving for roots of $A \omega^3 + \omega^2, C (\text{without} -\alpha_0) \omega - \alpha_0$ because $\frac{\partial}{\partial \omega}  \alpha_0 \log (\omega)= \alpha_0 \omega^{-1}$. The quadratic/cubic terms from revenue dominate the production cost in either case.
>
> > Currently this implies that operation costs hinge on the final performance reached after fine-tuning, not the baseline. Why does the generalist bear this cost?
>
> For a closed model that offers fine-tuning via API (e.g., GPT‑4o), the generalist handles inference costs and prices this into per-token API rates. This also ensures operation costs sum to a constant between players ($c_\omega \alpha_1$).
>
> > L128: "safety risk management for open models". This is a regulatory cost, but you are trying to provide an example for production costs.
>
> This example has been replaced.
>
> > Can you justify quadratic cost? Empirical fine-tuning curves often show super-linear returns for early steps.
>
> We want to make sure we’re interpreting your point correctly. Could you clarify why super-linear returns for early steps contradicts quadratic cost scaling? Our understanding is that “super-linear returns [for model performance] for early steps” is captured by a curve like $\alpha_0 = \text{cost}^{1/k}$, where initial investments get much more mileage for performance. Our quadratic cost claim describes the relationship $\text{cost} = \alpha_0^{k}$, where marginal cost for each unit of improvement gets more expensive. These are the same equations. Is it possible that we’re talking about the same relationship, just with flipped axes?
>
> > L157: "higher openness increases production costs". There is ample evidence showing this to be false. Projects like YOLO, vLLM, etc. have benefited drastically from open-source contributions. Outside of ML, look at Docker, Redis, etc. Even within foundation models, improvements to Gemma have led to improvements in Gemini.
>
> The line has been changed to clarify that higher openness increases production costs only for the *particular product* being offered at higher openness. We capture the benefits of open-source contributions through revenue rather than reduced production costs -- i.e., when an open model is improved by specialists to a higher level, $G$ recoups the increased production costs through higher revenue ($r(\alpha_1)$) because cheaper model access encourages specialists to invest more in $\alpha_1$.
> # Checklist
> We can upload the code as Supplementary Material. CPU runtime depends on the granularity of simulations. A simulation to produce a set of plots like those in Figures 4 and 5 with a sweep over 100 penalties, 100 thresholds, and 100 $\delta$ values (100,000 parameter combinations) executes in 317.362s / ~5 min on a single CPU. Similarly:
>
> * 150 $p$, 150 $\theta$, 100 $\delta \rightarrow$ 717.484s / ~12 min
> * 200 $p$, 200 $\theta$, 100 $\delta \rightarrow$ 1470.59s / ~24 min
>
> We were able to produce smooth plots at 200 x 200 x 100 granularity. For each plot in Appx. F, we used a sweep over 200 $\epsilon$, 200 $\alpha_0$, and 100 $\delta$, which executes in 1216.88s / ~20 min on a single CPU. We did not optimize for efficiency. Item 8 in the checklist has been updated with approximate lengths of runs and # of CPUs needed.

---

> > ### Comment · Reviewer_Rs1J · 2025-08-02
> >
> > Thank you for the detailed rebuttal.
> >
> > I think you've addressed almost all of my concerns. I have a few remaining comments:
> >
> > 1. "high vs. low performance is determined relative to the _current_ generation of models." How are you defining generations? Performance and generations are not cleanly separable insofar that the reasoning "generation" of models can perform worse than their non-reasoning counterparts, and vice versa. I think your paper could be greatly strengthened if you could propose a way to formalize the hand waviness around performance. I think this can be maintained for future work, however.
> > 2. "a closed model that offers fine-tuning via API" -> it would be great if you could specifically make this point in the paper (unless I've missed it somewhere). Right now I understand fine-tuning as anyone can fine-tune downstream on their own compute, not just via the generalist's providings. Are you sure that this simplification does not affect your modeling?
> > 3. "super-linear returns for early steps contradicts quadratic cost scaling" I think you're right, we are talking about the same relationship.

---

> > > ### Comment · Reviewer_Rs1J · 2025-08-02
> > >
> > > Also, please do provide your code if/when you release this work! It may be trivial to you, but it would drastically shorten the time it would take for anyone to follow up on your work :)

---

> > > ### Author Response · Authors · 2025-08-04
> > >
> > > > "a closed model that offers fine-tuning via API" -> it would be great if you could specifically make this point in the paper (unless I've missed it somewhere).
> > >
> > > Thanks for the suggestion; we’ve added this point to the explanation of $\phi_G$.
> > >
> > > > Right now I understand fine-tuning as anyone can fine-tune downstream on their own compute, not just via the generalist's providings. Are you sure that this simplification does not affect your modeling?
> > >
> > > This seems correct for models which at least have downloadable weights. If $\omega$ is high enough, $D$ will incur most of the operation costs by fine-tuning. When the model is mostly closed, we were thinking of two cases:
> > >
> > > * The closed model provider offers fine-tuning services, but this fine-tuning occurs on the generalist’s infrastructure (e.g., OpenAI’s fine-tuning APIs). This is the case we explained above.
> > > * $\omega$ is extremely low, so $D$ must choose $\alpha_1 = \alpha_0$. This is because $D$’s production costs ($(\alpha_1 - \alpha_0)^2 {\omega}^{-1}$) go to infinity as $\omega \rightarrow 0$ if $D$ tries to improve the model to $\alpha_1 > \alpha_0$. In this case, the distinction between $\alpha_1$ vs. $\alpha_0$ in $G$’s operation costs becomes irrelevant because $\alpha_1 = \alpha_0$.
> > >
> > > To verify that this doesn’t affect our modeling, we changed $G$’s operation cost from $c_\omega \alpha_1 (1-\omega)$ to $c_\omega \alpha_0 (1-\omega)$. The coefficients in $G$’s best response change from:
> > > * $A = -\frac{3}{2}  (\epsilon - \delta + c_\omega) (\delta + c_\omega)$ to $A = - \frac{(\delta + c_\omega)(\epsilon - \delta)}{2}$,
> > > * $B = (\epsilon - \delta + c_\omega)(1-\delta) + (c_\omega - \delta)(\delta + c_\omega)$ to $B = \frac{(\epsilon - \delta)(1 - \delta)}{2}$,
> > > * $C = \alpha_0 (\epsilon - \delta - 1 + c_\omega) + \frac{(\epsilon - c_\omega)(1-\delta)}{2}$ to $C=\alpha_0 (\epsilon - \delta - 1 + c_\omega)$.
> > >
> > > We compared plots for both versions at $c_\omega \in$ {$0.01, 0.1, 0.5, 1$} and $\alpha_0 \in$ {$0.1, 1$}. The plots are visually indistinguishable, with matching scales, abstain regions, and shading. For both versions, both players also completely abstain when $c_\omega \geq 1$.
> > >
> > > > How are you defining generations? Performance and generations are not cleanly separable insofar that the reasoning "generation" of models can perform worse than their non-reasoning counterparts, and vice versa. I think your paper could be greatly strengthened if you could propose a way to formalize the hand waviness around performance. I think this can be maintained for future work, however.
> > >
> > > We were using “generations” to loosely refer to release dates, but DeepSeek v3’s reasoning-focused post-training may explain why it looks like an outlier among open-weights models in Figure 3(b). Our paper assumes performance is 1D and models can be linearly ranked. However, the $\alpha_0 / \alpha_1$ parameters could be extended to multidimensional variables to capture when no single model consistently dominates across all dimensions (reasoning, general tasks, efficiency, etc.). This preserves a formal measure of performance, but it would significantly complicate the closed-form derivations. For example, $\frac{\partial U_D}{\partial \alpha_1}$ would become an $n$-dimensional vector $\nabla_{\alpha_1} U_D$.

---

### Official Review · Reviewer_t96V · 2025-07-01

**Clarity:** 3
**Significance:** 4
**Originality:** 4
**Rating:** 5
**Confidence:** 3

**Summary:**

The paper presents an economic model that tries to capture the economics of open sourcing foundational models for those who build them (the generalists) and those who finetunes these into specialized models (the domain specialists) given that the government enforces some form of regulation.

The results characterize market equilibria under various openness policies and present an optimal range of regulatory penalties and open-source thresholds as a function of the foundation model performance.

The model seeks to provide theoretical foundations for AI governance decisions around openness and enables evaluation and refinement of practical open-source policies, and the discussion concludes with three implications of the analysis.

**Questions:**

* Regarding the implications in 5.1: Would it not be better to articulate all three of them as advises to the regulatory bodies, as is the case with the second one? For example, the first one could be formulated as something like “Regulators should adopt tailored approaches according to the maturity of the technology”? I think the results would be more accessible to regulatory bodies, who might not be as easily sold on pareto improvements.
* Why is there not a constant term specifying the cost in the production cost term $\alpha_0\omega$? As I read the term, both $\alpha_0$ and $\omega$ are rates. When they are added together with operation and regulatory which both contains constants ($c_w$ and $p$), then this would not yield the correct result. Am I wrong? What is it that I do not understand?
* Are there other parameters that could lead to the same behavior as described in Figure 3 than those that you give?
* Regarding negative examples: Are there real-world examples, such as the one given in Figure 3, that are not described by the model. If yes, then why does it not capture this?
* As far as I understand, the model only captures a single $G$ and a single $D$. How would many $G$ to many $D$ change the analysis. Would it be a worthwhile analysis? Is this future work?

**Ethical Concerns:**

["NO or VERY MINOR ethics concerns only"]

**Final Justification:**

My main concern was related to clarity, which the authors state that they have resolved. Also, their responses to the questions and especially those related to limitations of the method seem nice.

My concerns have been resolved, and I think this paper should be accepted to the conference.

**Limitations:**

I think this is sufficiently addressed.

**Paper Formatting Concerns:**

I have no concerns regarding the formatting.

**Quality:**

4

**Strengths And Weaknesses:**

## Strengths
* The topic is novel, and the potential impact could be high given that it could help governments in their understanding of how to regulate foundational models. Such regulations are already in place in several countries, including EU, US, and Singapore, as explained in the introduction.
* The model is simple but still captures the main drivers, and the analysis seems to me – at least – to be sound. I especially like the discussion where the central findings are discussed, although I have some suggestions for how to improve them, see below.
* The paper is very well structured and generally well written.
* I really like the explanations of the costs in the equations on line 125.
* I like how you show how the model explains the real-world example in Figure 3.

## Weaknesses
* My main gripe with the paper is the writing but not all of it. I found especially the parts where the model was introduced in detail to be hard to read because variables were not properly introduced before using them. The introduction came at a later point, but this made me very confused when reading through it the first time. I do not think it will take a lot of effort to fix this. Quite the opposite, I think it is quite easy. Examples are $\epsilon$, $p$ and $c_W$. Utility $U$ and cost ($\phi$) are shown in Table 1 but not mentioned in the text. I would move Table 1 later in section 2.3 and introduced all variables in the text before using them, maybe even in section 2.2.
* I would like a discussion on negative examples. Which scenarios will the model fail to capture?

## Minor comments
* Line 23: OSS describes publicly available source code with no restrictions on use … This is not fully correct. Many OSS licenses do not restrict use, but some do, especially those that have twin licenses where academic and non-commercial use is OK but commercial use is restricted. Such licenses are not uncommon even in academia, especially in the fields of AI and ML.
* Fully open models, as described in lines 26 and 27, should also share the code used for training and inference/reasoning. If not, they are not reproducible (which should be the main goal of open-sourcing foundation models). See for example (Gundersen, 2021) for a definition of reproducibility that requires code and data.
 * Instead of using the term “industry developers” on line 29, which targets individuals and might therefore be much less coherent, I suggest using the term “software industry” or “industry” as this indicates companies, which is how I read the sentence anyhow.
* The comment on $\epsilon$ on line 117 confuses more than it helps, especially as the variable is not introduced..
* The caption of Figure 4 is very parameter description heavy. A higher level-explanation would be nice. If possible.

## Reference
Gundersen, O. E. (2021). The fundamental principles of reproducibility. Philosophical Transactions of the Royal Society A, 379(2197), 20200210.

---

> ### Author Rebuttal · Authors · 2025-07-25
>
> We appreciate the positive review and have addressed your questions below.
>
> # Questions
> > Are there other parameters that could lead to the same behavior as described in Figure 3 than those that you give?
>
> Increasing $\alpha_0$ under no penalty always leads to more closed behavior as long as $\epsilon < 1$ and $c_\omega < 1$. For example, the openness trends in Figure 2 and Appendix F continue for higher values of $\alpha_0$ (we tested at $\alpha_0 = 10$ and $\alpha_0 = 100$). We require $\epsilon < 1$ because when reputational benefits always match or exceed closed revenue ($\epsilon=1$), $G$ will just open the model fully.
>
> The behavior in Figure 3 also requires that  $c_\omega < 1$ because if operation costs exceed any revenue $G$ can earn from model adoption, $G$ always fully opens the model or abstains at any performance level. $c_\omega < 1$ is a realistic assumption because developers can price the costs of inference into model offerings via API pricing or subscription charges.
>
> > As far as I understand, the model only captures a single $G$ and a single $D$. How would many $G$ to many $D$ change the analysis. Would it be a worthwhile analysis? Is this future work?
>
> Yes, this definitely seems like worthwhile analysis. We have noted modeling competition between multiple players as a future direction.
>
> **Extension to multiple $G$ with varying $\alpha_0$ levels.** This can only be done via simulation. In the first stage of backward induction, $D$ would need to solve multiple optimization problems and choose the $G$ that maximizes their utility payoff (potentially with switching costs). Both $\alpha_0$ and $c_\omega$ would be specific to each $G$.
>
> **Extension to multiple $D$.** $G$’s revenue would be the sum of all $\alpha_1$ for the $D$ who adopt their model. However, this extension is trickier because it requires modelling multiple $D$ with different valuations and priorities; e.g., one specialist might prefer cheaper operation costs if inference demand is high, while another tolerates high operation costs for better model accuracy due to domain-specific requirements. Future work could model the strategic dynamics of $G$ differentiating their product along different dimensions -- higher model openness vs. higher performance -- to target a specific subset of specialists (see: Giarratana & Fosfuri, “Product Strategies and Survival in Schumpeterian Environments: Evidence from the US Security Software Industry”).
>
> Modeling multiple $G$ and $D$ would require solving a separate bargain for each ($G, D$) pairing. Since we identify $\delta^*$ via grid search over the interval [0,1] and there are $G/D$-specific cost functions with multiple players, this would be intractable as add-on analysis for the paper.
>
> > Why is there not a constant term specifying the cost in the production cost term $\alpha_0\omega$? As I read the term, both $\alpha_0$ and $\omega$ are rates. When they are added together with operation and regulatory which both contains constants ($c_w$ and $p$), then this would not yield the correct result. Am I wrong? What is it that I do not understand?
>
> It’s a good idea to include the constant terms more explicitly. We had analyzed the version where constants for production and regulatory costs are normalized to 1 and the $c_\omega$ constant compares between operation costs and revenue. However, we realize we can provide more general results where these constants are any positive number. We have now included all constant terms into closed-form solutions and our numerical simulations. An important empirical direction would be estimating appropriate values for the two constants based on how regulatory requirements map to dollar value penalties, how documentation costs correspond to inference costs, etc.
>
> # Clarifications
>
> > Many OSS licenses do not restrict use, but some do, especially those that have twin licenses where academic and non-commercial use is OK but commercial use is restricted.
>
> These distinctions seem significant, so we have updated L23 to reflect nuances in OSS licensing. We had originally written “no use restrictions” based on the Open Source Initiative definition of OSS.
>
> > Fully open models, as described in lines 26 and 27, should also share the code used for training and inference/reasoning. If not, they are not reproducible (which should be the main goal of open-sourcing foundation models).
>
> This has been noted in the text, with supporting examples drawn from Gundersen (2021).
>
> # Weaknesses
> We have revised Section 2 for clarity and changed Section 5 to rearticulate findings as high-level considerations for regulatory bodies without referencing player utilities or Pareto improvements.
>
> > I would like a discussion on negative examples. Which scenarios will the model fail to capture?
>
> See Appendix C for our justifications for omitting considerations like deployment scale. To more explicitly discuss negative examples, the discussion in Section 5.2 has been revised with more examples: “The two-player framework also overlooks important social utility considerations. While our analysis assumes that higher model openness and performance are generally beneficial, a regulator must consider safety risks associated with higher performance or market-wide loss scenarios, such as possible intermediate levels of openness where consumers may be worse off. This complexity suggests that defining openness as a multidimensional property may be valuable to capture factors beyond simple model access, such as documentation and explainability of model outputs. Additionally, modeling multi-firm competition between a closed-source incumbent and multiple generalists can reveal when established firms pivot to open-source strategies and how timing of market entry affects whether entrants adopt open-source approaches.”

---

> > ### Comment · Reviewer_t96V · 2025-08-06
> > **Thanks for detailed answers to my comments and questions**
> >
> > Your comments are much appreciated, and you have addressed my concerns.

---

### Decision · Program_Chairs · 2025-09-17

**Decision:**

Accept (poster)

**Comment:**

This paper touches upon an important subject (AI governance and the expected impacts of regulation).

After a fruitful discussion with authors, all reviewers agreed that this paper was worth accepting. Therefore I recommend acceptance.